# Unlocking the Essence of Beauty: Advanced Aesthetic Reasoning with Relative-Absolute Policy Optimization

**Boyang Liu**[1,4*]  **Yifan Hu**[3,*]  **Senjie Jin**[1,4,*]  **Shihan Dou**[1]  **Gonglei Shi**[4]
**Jie Shao**[4,†]  **Tao Gui**[1,2,‡]  **Xuanjing Huang**[1,2,‡]

[1]Fudan University  [2]Shanghai Collaborative Innovation Center of Intelligent Visual Computing
[3]Tsinghua University  [4]Bytedance
{boyangliu25,sjjin24}@m.fudan.edu.cn  tgui@fudan.edu.cn

## Abstract

Advancements in multimodal large language models (MLLMs) demonstrate strong potential across various domains, with expanding applications in human-centered Image Aesthetic Assessment (IAA) through high-level cross modal understanding capacity. Compared to data-intensive supervised fine-tuning (SFT), the efficient and scalable reinforcement learning (RL) is a promising alternative for IAA. However, directly applying RL to IAA fails to inspire reasoning patterns without prior aesthetic knowledge, and modeling an appropriate reward proxy to evaluate aesthetic scores is intricate. To this end, we propose Aes-R1, a comprehensive IAA framework. Concretely, Aes-R1 integrates a data pipeline, AesCoT, to construct and filter high-quality aesthetic reasoning data used for cold-start. After teaching the model to generate structured explanations before scoring, we employ the Relative-Absolute Policy Optimization (RAPO), a novel RL algorithm that jointly optimizes absolute score regression and relative ranking order, improving both per-image accuracy and cross-image preference judgments. Extensive experiments demonstrate that Aes-R1 improves the backbone's average PLCC/SRCC by 47.9%/34.8%, surpassing state-of-the-art baselines of similar training size. More ablation studies validate Aes-R1's robust generalization in out-of-distribution scenarios with limited supervision.[1]

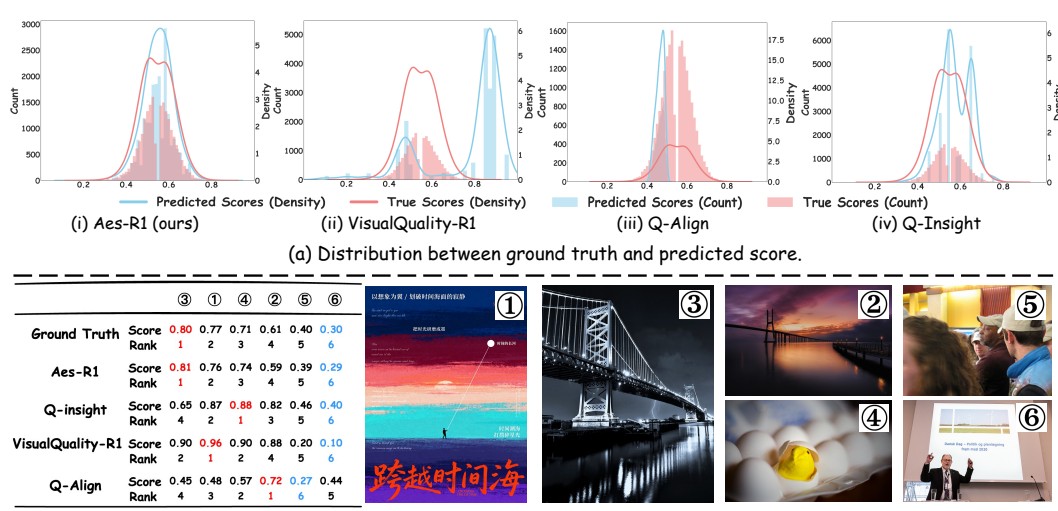

(a) Distribution between ground truth and predicted score.

(b) Comparison of various methods.

Figure 1: (a) Distributions of ground truth and predictions on the AVA test set. Aes-R1 achieves the closest alignment with the ground truth compared to baseline methods. (b) The case study shows that Aes-R1 surpasses other methods, which suffer from prediction errors and limited ordinal discernment, based on the relative-absolute reward design.

---

*Equal Contribution  † Project lead  ‡ Corresponding author
[1]Our code is available at https://github.com/ssssmark/AesR1

# 1 INTRODUCTION

Recent advances in multimodal large language models (MLLMs) (Qwen et al., 2025; OpenAI et al., 2024; Team et al., 2025) have shown remarkable potential in image assessment (Xu et al., 2023; Wu et al., 2023b). To date, most efforts center on image quality assessment (IQA) (Wu et al., 2024; 2023a; You et al., 2025), which measures images' sharpness and clarity. As the development of creative design and social media recommendations, there is an emerging demand for reliable metrics that go far beyond pixel-level fidelity and some researchers have paid attention to image aesthetic assessment (IAA) (Huang et al., 2024; Zhou et al., 2024), leveraging the cross-modal understanding capabilities of MLLMs to capture high-level features such as texture, color harmony and emotional impact, broadening vision tasks to the interpretable and human-centered aesthetic modeling.

Current IAA methods generally adopt supervised fine-tuning (SFT) on score-based targets (Wu et al., 2023a; Cao et al., 2025). While such implicit reasoning restricts MLLMs from aligning visual elements with multidimensional aesthetic criteria and lacks explainability in practice (Wu et al., 2025; Li et al., 2025), some propose generating reasoning prior to the answer, supporting the integration of fine-grained descriptive feedback with aesthetic scores (Li et al., 2025). Nonetheless, existing resources (He et al., 2022; Yang et al., 2022) primarily consist of score-based data, and artist-level rationales necessitate expensive expert annotation. Moreover, as presented in Tab. 3, extensive SFT causes the model to learn bias and overfit on curated dataset, rapidly reducing entropy and hindering further optimization, making it costly and difficult to scale.

Reinforcement learning (RL) is an effective alternative and has shown strong efficiency and generalization in goal-driven tasks (Chu et al., 2025a; Chen et al., 2025a). However, in our pilot study, directly applying RL to IAA exposes several challenges. 1) Although achieving high performance (Tab. 3, SFT 0 epoch, RAPO), the end-to-end RL fails to activate aesthetic reasoning patterns in the backbone due to the absence of corresponding corpus pre-training (Gandhi et al., 2025; Ji et al., 2025), resulting in the poor explanatory rationales as shown in Fig. 6 (AesR1-Zero). 2) Modeling an appropriate reward proxy to evaluate aesthetic score is intricate since the absence of unified criteria for aesthetic preferences. We reproduce the related works on AVA (Murray et al., 2012) and find that, although VisualQuality-R1 (ii) (Li et al., 2025) successfully separates high- and low-quality images using the rank reward, it fails to calibrate scores, while Q-insight (iv) (Wu et al., 2025) suffers from peak mismatch when optimizing with a scalar reward, as presented in Fig. 1.

Integrating aesthetic priors and an effective reward mechanism is crucial for IAA in RL optimization. Inspired by Deepseek-R1 (Guo et al., 2025a), we find that cold-start bootstrapping with thousands of high quality reasoning data effectively induces robust aesthetic analysis patterns (Gandhi et al., 2025; Shao et al., 2025). Following this, RL transforms IAA into a data efficient self-learning process, primarily guided by the reward signal. Building on the results in Fig. 1, we observe that human aesthetic judgment is inherently context-dependent besides absolute quality evaluation, as also noted by Immanuel Kant's assertion that "*there can be no objective rule of taste which shall determine by means of concepts what is beautiful*" (Kant, 1790). Therefore, we argue the IAA reward should comprise two complementary facets: a relative dimension that ranks images by comparative appeal and an absolute dimension that assesses intrinsic aesthetic merit, thereby simultaneously enforces ordinal consistency and calibrates absolute aesthetic scores.

Building on the above insights, we propose **Aes-R1** framework. We first construct an automatic data pipeline **AesCoT**, which effectively generates high-quality aesthetic reasoning data along five dimensions based on original image-score pairs. Warming up on this data equips MLLMs with aesthetic cognitive behaviors that generating reliable explanations before the aesthetic scores. We then utilize Relative-Absolute Policy Optimization (RAPO), a novel RL algorithm that optimizes both absolute and relative aesthetic preferences. Specifically, the model's gradient updates are simultaneously steered by the absolute-error reward, calibrating aesthetic score constraints and relative-rank reward, aligning pairwise ranking consistency, which greatly improves effectiveness and stability in the exploration and exploitation process. Empirically, trained with only 15K data, Aes-R1 surpasses the state-of-the-art baselines of comparable size with the improvements on backbones by 47.9%/34.8% PLCC/SRCC. The distribution alignment experiment (see Fig. 1, a) and reward ablations (see Sec. 4.3) demonstrate the effectiveness of our RAPO as well as generalization in the OOD setting (see Sec. 4.2). We further analyze the training recipe of SFT and RL from the entropy perspective (Cui et al., 2025b) (see Sec. 4.3).

In summary, our main contributions can be summarized as follows:

❶ We construct the effective and automatic data pipeline, **AesCoT**, which builds high-quality aesthetic reasoning data along five dimensions based on original image-score pairs, significantly reducing labor cost. We will release our **AesCoT-3K** and **AesCoT-10K** to facilitate future research in IAA.

❷ We develop a two-stage training pipeline, **Aes-R1**, based on Relative-Absolute Policy Optimization (RAPO) that jointly optimizes absolute score regression and relative ranking, enabling balanced modeling of intrinsic merit and comparative preference in IAA.

❸ We conducted extensive experiments to demonstrate the performance, robustness and generalization of Aes-R1, which achieves state-of-the-art performance across multiple IAA benchmarks with only 15K training samples. We also analyze the reward design and training recipe for SFT and RL, offering valuable insights.

## 2 RELATED WORK

**Score-Based Image Aesthetic Assessment.** Research on Image Aesthetic Assessment (IAA) has progressed from rule-based and handcrafted pipelines (Bhattacharya et al., 2010; Mavridaki & Mezaris, 2015; Mittal et al., 2013; 2012) to deep models supervised by human-annotated mean opinion scores (MOSs). Deep learning IAA models (Talebi & Milanfar, 2018; Ke et al., 2021; Ghosal & Smolic, 2022; He et al., 2022; Behrad et al., 2025) became dominant after large-scale aesthetic datasets (Murray et al., 2012; He et al., 2022; Ren et al., 2017; Yang et al., 2022; Kong et al., 2016) enabled direct pixel-level representation learning via powerful networks. More recently, pretrained MLLMs have demonstrated remarkable cross-modal aesthetic perception and evaluation through visual-language pretraining (Ke et al., 2023), prompt engineering (Zhu et al., 2024; Sheng et al., 2025), and supervised fine-tuning (Zhang et al., 2023; Cao et al., 2025; Liao et al., 2025; Zhou et al., 2024). Nevertheless, score-only supervision without explicit linguistic rationales limits aesthetic understanding, weakens generalization and requires redundant dataset-specific training.

**Reinforcement Learning-Enhanced Model-as-a-Judge.** Model-as-a-Judge approach serves as a novel and broadly applicable paradigm for addressing open-ended evaluation problems in the era of Large Models (Gu et al., 2025). Prior works on Model-as-a-Judge (Zheng et al., 2023; Zhu et al., 2025; Chen et al., 2024; Lee et al., 2024) as well as the Generative Reward Models (GRMs) (Mahan et al., 2024) highlight that reasoning step-by-step before verdict improves the stability and consistency (Kim et al., 2024; Saha et al., 2025). Capitalizing on recent advances in post-training (Liu et al., 2025; Yu et al., 2025), various large reasoning judge models have emerged (Liu et al., 2025; Chen et al., 2025b; Whitehouse et al., 2025) and have become popular fro automated assessment. Extending to image evaluation, researchers move beyond scalar scoring (Wu et al., 2023a; You et al., 2025; Zhou et al., 2024) by applying reinforcement learning to image quality assessment via step-by-step reasoning (Li et al., 2025; Wu et al., 2025). However, these methods are confined to either binary outcome supervision or rank supervision and have yet to be extended to aesthetic assessment, which demands finer-grained perceptual and semantic understanding capability.

## 3 METHODOLOGY

There is indeed an absence of a principled visual reasoning framework for IAA, which contributes to the scarcity of high-quality aesthetic reasoning data and the misalignment of single-objective optimization with nuanced human preferences. To address this, we propose our Aes-R1 framework. Specifically, we introduce (i) AesCoT, an aesthetic reasoning data construction pipeline that automatically curates diverse, reliable annotations with multi-dimensional aesthetic rationales. (ii) A two-stage training pipeline based on Relative-Absolute Policy Optimization (RAPO) that jointly optimizes pairwise preferences and calibrated absolute scores under a unified objective.

### 3.1 PROBLEM FORMULATION

Given an image dataset that contains $M$ image-score pairs $\mathcal{D} = \{(\mathcal{I}_i, s_i), i \in [1, M]\}$, considering a pretrained MLLM with parameter $\theta$ and policy $\pi_\theta(\cdot|\mathcal{I}, \mathcal{P})$, we aim to fine-tune the model for ideal

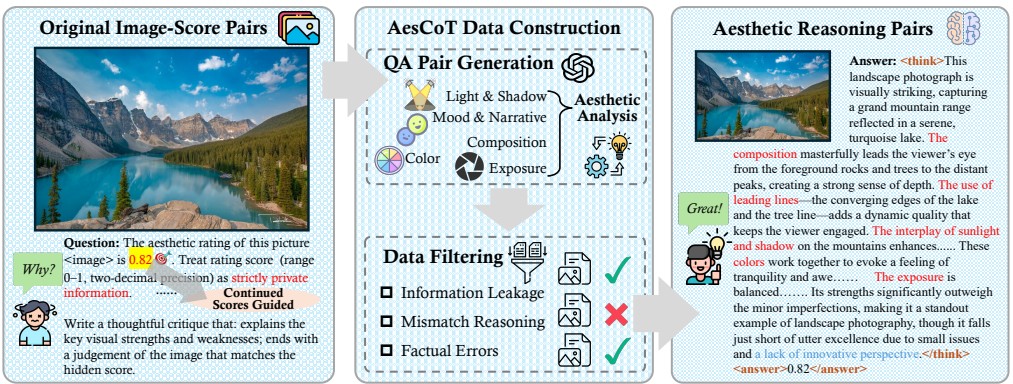

Figure 2: Overview of AesCoT construction pipeline. Starting from original image-score pairs, we mask the continuous aesthetic score and prompt experts to produce CoT critiques along five aesthetic dimensions. Automated checks and human audits then remove any score leakage, reasoning–score mismatch, or factual errors, yielding high-quality, interpretable multimodal aesthetic reasoning data.

aesthetic reasoning that provides the comprehensive explanation $c$ and an overall aesthetic score $s$ for every image in dataset $\mathcal{I}$ under prompt $\mathcal{P}$:

$$\tau = (c, s) \sim \pi_\theta(\cdot | \mathcal{I}, \mathcal{P}) \tag{1}$$

We utilize policy gradient methods (Sutton et al., 1999) to optimize the MLLM by maximizing the expected cumulative reward $R(\tau)$ on sampled aesthetic reasoning trajectories $\tau$: $\mathcal{J}(\theta) = \mathbb{E}_{\tau \sim \pi_\theta(\cdot|\mathcal{I},\mathcal{P})}[R(\tau)]$. By performing gradient ascent on $\mathcal{J}(\theta)$:

$$\nabla_\theta \mathcal{J}(\theta) = \mathbb{E}_{\tau \sim \pi_\theta(\cdot|\mathcal{I},\mathcal{P})}\left[ R(\tau) \sum_{t=1}^{T} \nabla_\theta \log \pi_\theta(a_t \mid s_t) \right], \tag{2}$$

where $a_t$ is selected from the vocabulary space $\mathcal{V}$ and $s_t = (\mathcal{I}, \mathcal{P}, a_{<t})$ is the state at time step $t$.

## 3.2 AESCoT

Most existing datasets (e.g., AVA (Murray et al., 2012)) annotate image aesthetics using direct Mean Opinion Scores (MOSs). Chain-of-Thought (CoT) (Wei et al., 2022) has been shown to substantially enhance the performance and interpretability by breaking down the reasoning process into intermediate steps (Bi et al., 2025). However, high-quality aesthetic reasoning trajectories require artist-level curation and iterative refinement, making such data rare and costly to scale. Distillation from a strong model is an efficient and scalable approach (Guo et al., 2025a) and was validated in domains from math reasoning (Jie et al., 2023) to visual question answering (Shao et al., 2024a). Based on these, we propose AesCoT, to our knowledge, the first automatic pipeline for aesthetics reasoning data construction.

We display the construction pipeline of AesCoT (see Fig. 2). For every collected original image-score pair $(\mathcal{I}_i, s_i)$ in $\mathcal{D}$, we first intentionally conceal the continuous score from outputs and enable the powerful close-sourced MLLM to give aesthetic analysis over five aesthetic dimensions (see Appendix E.1) (light and shadow, mood and narrative, composition, color, exposure) that match the aesthetic levels, which are divided into bad (0-0.4), fair (0.4-0.7) and good (0.7-1.0) to emphasize aesthetic differences. Then we concatenate the aesthetic analysis with the ground truth score, obtaining $\mathcal{D}_{CoT} = \{(\mathcal{P}, \mathcal{I}_i, c_i, s_i), i \in [1, M]\}$ where $c_i$ is the reasoning trajectory towards $s_i$. Finally, we carefully filter the outputs to eliminate score leakage, reasoning inconsistencies, and other factual errors through automated checks and human audits, generating a reliable aesthetic reasoning dataset $\mathcal{D}_{AesCoT} = \mathcal{F}(\mathcal{D}_{CoT}) = \{(\mathcal{P}, \mathcal{I}_i, c_i, s_i) \mid \|\mathcal{E}_i\| = \mathbf{0}\}$, where $\mathcal{F}$ is the data filter process and $\mathcal{E}_i = (e_i^{\text{leak}}, e_i^{\text{align}}, e_i^{\text{fact}})$ represent different error types. Dataset details can be found in **??**.

## 3.3 RELATIVE-ABSOLUTE POLICY OPTIMIZATION (RAPO)

Aesthetic evaluation is inherently subjective, shaped by individual preferences, cultural experience, and contextual cues. In real-world settings, human judgments are typically expressed through com-

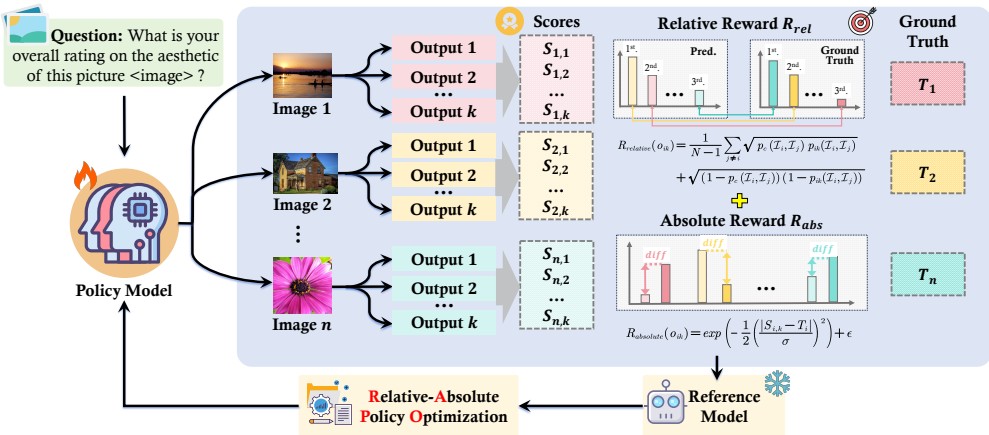

Figure 3: The training pipeline of Relative Absolute Policy Optimization (RAPO). Given a batch of training images, the policy model generates multiple outputs for aesthetic rating questions. RAPO computes both pairwise ranking based relative reward and score regression error based absolute reward for each output, then optimizes the policy model to align with human aesthetic preferences.

parative assessments, in addition to direct absolute scores, with judgments manifesting along two complementary axes: relative preference (which image is favored) and absolute magnitude (by how much). In view of this, we introduce Relative Absolute Policy Optimization (**RAPO**), a novel reinforcement learning algorithm that integrates relative assessment learning with absolute numerical optimization to better align model predictions with human aesthetic preferences.

Fig. 3 describes the training pipeline of RAPO. For a mini training batch $\{(\mathcal{I}_1, s_1), \ldots, (\mathcal{I}_N, s_N)\}$, where the training size is $N$, we follow GRPO (Shao et al., 2024b) to sample a group of $K$ outputs $\{o_{i1}, o_{i2}, \ldots, o_{iK}\}$ for every prompt-image input $(\mathcal{P}, \mathcal{I}_i)$ from the old policy $\pi_\theta^{old}$.

A robust and reliable reward signal is significant for aligning MLLMs with human aesthetic preferences. To this end, RAPO jointly optimizes the relative rank preference as well as the absolute score regression error.

**Relative Rank Reward.** Learning to rank is a crucial paradigm for modeling human preference (Burges et al., 2005; Xia et al., 2008), and has proven effective for perceptual quality and aesthetic assessment (Liu et al., 2017; Talebi & Milanfar, 2018). Following (Tsai et al., 2007; Sun et al., 2024; Wu et al., 2025; Zhang et al., 2023; You et al., 2025), our relative rank reward based on FRank is bounded, continuous, differentiable, and aligns directly with pairwise ranking consistency, and thus provides stable, precise signals for optimizing pairwise image aesthetic preference.

Specifically, we assume that the aesthetic score of an image follows the Gaussian distribution $s \sim \mathcal{N}(\mu, \sigma^2)$. Consequently, the score difference $s_i - s_j$ follows $\mathcal{N}(\mu_i - \mu_j, \sigma_i^2 + \sigma_j^2)$. Let $o_{ik}$ denote the $k$-th predicted score for image $i$, the pairwise comparison probability against image $j$ should be

$$p_{ik}(\mathcal{I}_i, \mathcal{I}_j) = \Phi\left(\frac{o_{ik} - \mu_j}{\sqrt{\sigma_i^2 + \sigma_j^2 + \gamma}}\right), i \neq j, \tag{3}$$

where $\Phi(\cdot)$ denotes the standard normal cumulative distribution function. We estimate $\mu_j$ as the mean of $k$ predicted scores of image $j$, and set a small positive constant $\gamma$ to avoid division-by-zero.

$$r_{rank}(o_{ik}) = \frac{1}{N-1} \sum_{j \neq i} \sqrt{p_c(\mathcal{I}_i, \mathcal{I}_j)\, p_{ik}(\mathcal{I}_i, \mathcal{I}_j)} + \sqrt{(1 - p_c(\mathcal{I}_i, \mathcal{I}_j))(1 - p_{ik}(\mathcal{I}_i, \mathcal{I}_j))}, \tag{4}$$

where $p_c$ is a binary preference label based on ground-truth MOSs of two images

$$p_c(\mathcal{I}_i, \mathcal{I}_j) = \begin{cases} 1, & s_i \geq s_j \\ 0, & s_i < s_j \end{cases}. \tag{5}$$

**Absolute Error Reward.** Although the rank reward reinforces the model to order images correctly, it leaves the calibrated scoring underconstrained. To fix this, we incorporate a continuous absolute-error reward that explicitly calibrates predicted scores toward the ground-truth MOSs, where $\sigma$ is a hyperparameter and $\epsilon$ is a small positive constant to avoid reward being zero.

$$r_{abs}(o_{ik}) = exp\left(-\frac{1}{2}\left(\frac{|o_{ik} - s_i|}{\sigma}\right)^2\right) + \epsilon. \tag{6}$$

The total reward function of RAPO is combined with relative rank reward and absolute error reward

$$r = r_{rank} + r_{abs}. \tag{7}$$

**Training Aes-R1 for Aesthetic Reasoning.** To fully inspire the aesthetic reasoning capability of the MLLM, we first leverage the constructed AesCoT dataset to fine tune the model as a cold start stage, optimizing the negative log-likelihood of the aesthetic reasoning trajectory and the final score

$$\mathcal{L}_{sft}(\theta) = \mathbb{E}_{(\mathcal{P},\mathcal{I},c,s)\sim\mathcal{D}_{CoT}}\left[-\log\pi_\theta(c, s|\mathcal{P},\mathcal{I})\right]. \tag{8}$$

For the prompt-image pair $(\mathcal{P}, \mathcal{I}_i)$, we calculate the rewards of all sampled outputs, yielding rewards $R_i = \{r_1, r_2, \ldots, r_K\}$. The advantage $\hat{A}_{k,t}$ for each token time step $t$ in the $k$-th output is given by subtracting the group average and dividing by the standard deviation

$$\hat{A}_{k,t} = \frac{r_k - \mu(R_i)}{\sigma(R_i)}. \tag{9}$$

Following DAPO (Yu et al., 2025), we adopt a higher clipping threshold and reduce the KL penalty terms. RAPO optimizes the policy by maximizing the following objective

$$\mathcal{J}_{\text{RAPO}}(\theta) = \mathbb{E}_{(\mathcal{P},\mathcal{I})\sim Q,\, o_i\sim\pi_\theta^{\text{old}}(\cdot|\mathcal{P},\mathcal{I})}$$

$$\left[\frac{1}{K}\sum_{k=1}^{K}\frac{1}{|o_{ik}|}\sum_{t=1}^{|o_{ik}|}\min\left(r_{k,t}(\theta)\hat{A}_{k,t},\, \text{clip}\left(r_{k,t}(\theta), 1-\epsilon_{\text{low}}, 1+\epsilon_{\text{high}}\right)\hat{A}_{k,t}\right) - \beta\,\mathcal{D}_{\text{KL}}(\pi_\theta \,\|\, \pi_{\text{ref}})\right], \tag{10}$$

where $\pi_{\text{ref}}$ denotes the reference policy of pretrained model, and the probability ratio is defined as

$$r_{k,t} = \frac{\pi_\theta(o_{k,t}|\mathcal{P},\mathcal{I},o_{k,<t})}{\pi_\theta^{old}(o_{k,t}|\mathcal{P},\mathcal{I},o_{k,<t})}, \tag{11}$$

and the KL penalty is approximated by

$$\mathcal{D}_{\text{KL}}(\pi_\theta \,\|\, \pi_{\text{ref}}) = \frac{\pi_{\text{ref}}(o_{k,t}|\mathcal{P},\mathcal{I},o_{k,<t})}{\pi_\theta(o_{k,t}|\mathcal{P},\mathcal{I},o_{k,<t})} - \log\frac{\pi_{\text{ref}}(o_{k,t}|\mathcal{P},\mathcal{I},o_{k,<t})}{\pi_\theta(o_{k,t}|\mathcal{P},\mathcal{I},o_{k,<t})} - 1. \tag{12}$$

## 4 EXPERIMENT

### 4.1 EXPERIMENT SETUP

**Dataset and Metrics.** We adopt five open-source general image aesthetic assessment datasets: **TAD66K** (He et al., 2022), **AVA** (Murray et al., 2012), **FLICKR-AES** (Ren et al., 2017), **PARA** (Yang et al., 2022), and **AADB** (Kong et al., 2016). The model is trained on TAD66K, AVA, and Flickr-aes, with PARA and AADB serving as out-of-distribution (OOD) datasets. Mean Opinion Scores (MOSs) across these datasets are normalized into the range $[0, 1]$. For evaluation, model performance is quantified using the Pearson linear correlation coefficient (**PLCC**) and the Spearman rank-order correlation coefficient (**SRCC**) between predicted scores and ground-truth annotations. The implementation details can be found in Appendix C.

**Baselines and Implementation.** We compare our method with : (1) popular LLMs with various sizes, capacities and reasoning modes, including **Qwen-2.5-VL** series (Qwen et al., 2025) , **Seed-1.5-VL-thinking** (Guo et al., 2025b), **GPT4o** (OpenAI et al., 2024) and **GPT4.1** (OpenAI, 2025); (2) HandCrafted methods like **NIQE** (Mittal et al., 2013) and **BRISQUE** (Mittal et al., 2012); (3) Deep Learning based methods including **MUSIQ** (Ke et al., 2021) and **NIMA** (Talebi & Milanfar, 2018); (4) MLLM based methods including **VILA** (Ke et al., 2023), **Laion-Aes** (Schuhmann et al., 2022), **Q-Align** (Wu et al., 2023a), **ArtiMuse** (Cao et al., 2025), **DeQA-Score** (You et al., 2025), **Q-Insight** (Li et al., 2025) and **VisualQuality-R1** (Wu et al., 2025). The implementation details is in Appendix C.

| Model | TAD66K | | AVA | | FLICKR | | PARA | | AADB | | Average | |
|---|---|---|---|---|---|---|---|---|---|---|---|---|
| | PLCC | SRCC | PLCC | SRCC | PLCC | SRCC | PLCC | SRCC | PLCC | SRCC | PLCC | SRCC |
| *Vanilla MLLM* | | | | | | | | | | | | |
| Qwen2.5-VL-7B | 0.2282 | 0.2242 | 0.3518 | 0.3684 | 0.5179 | 0.5696 | 0.5966 | 0.6252 | 0.4480 | 0.5076 | 0.4285 | 0.4589 |
| Qwen2.5-VL-32B | 0.2300 | 0.2715 | 0.3893 | 0.4170 | 0.5800 | 0.6527 | 0.7176 | 0.7118 | 0.5060 | 0.5138 | 0.4846 | 0.5134 |
| Seed1.5-VL-thinking | 0.2908 | 0.2974 | 0.4533 | 0.4871 | 0.5837 | 0.6374 | 0.6134 | 0.6364 | 0.5279 | **0.5577** | 0.4938 | 0.5232 |
| GPT4o-2024-08-06 | 0.2686 | 0.2979 | 0.4687 | 0.4939 | 0.5469 | 0.5507 | 0.6916 | 0.6719 | 0.5209 | 0.5314 | 0.4993 | 0.5092 |
| GPT-4.1-2025-04-14 | 0.2707 | 0.3298 | 0.4846 | 0.5594 | 0.5909 | 0.6062 | 0.7290 | 0.7203 | 0.5102 | 0.5296 | 0.5171 | 0.5491 |
| *Hand-Crafted* | | | | | | | | | | | | |
| NIQE (2013) | 0.1332 | 0.0811 | 0.1226 | 0.0448 | 0.1024 | 0.0487 | 0.3546 | 0.2478 | 0.0698 | 0.0566 | 0.1565 | 0.0958 |
| BRISQUE (2012) | 0.0294 | 0.0243 | 0.0310 | 0.0360 | 0.0830 | 0.0910 | 0.1288 | 0.0783 | 0.0565 | 0.0558 | 0.0657 | 0.0571 |
| *Deep-Learning Based* | | | | | | | | | | | | |
| NIMA (2018) | 0.3885 | 0.3654 | 0.6120 | 0.6361 | 0.5130 | 0.4796 | 0.5868 | 0.5709 | 0.3886 | 0.3904 | 0.4978 | 0.4885 |
| *MLLM Based* | | | | | | | | | | | | |
| Q-Align* (2023a) | 0.3498 | 0.3627 | 0.5215 | 0.5407 | 0.6231 | 0.6473 | 0.6181 | 0.6262 | 0.4474 | 0.4508 | 0.5120 | 0.5255 |
| DeQA-Score* (2025) | 0.3985 | 0.3885 | 0.5718 | 0.5896 | 0.7057 | 0.6889 | 0.7234 | 0.6796 | 0.4859 | 0.4948 | 0.5771 | 0.5663 |
| Q-Insight* (2025) | 0.3980 | 0.3886 | 0.5964 | 0.5898 | 0.7012 | 0.6769 | 0.7745 | 0.7428 | 0.5069 | 0.5184 | 0.5954 | 0.5813 |
| VisualQuality-R1* (2025) | 0.3082 | 0.3915 | 0.4407 | 0.6195 | 0.5524 | 0.6769 | 0.5376 | 0.7507 | 0.3754 | 0.5363 | 0.4429 | 0.5930 |
| Aes-R1 | **0.4513** | **0.4248** | **0.6702** | **0.6619** | **0.7243** | **0.6973** | **0.7842** | **0.7666** | **0.5386** | 0.5423 | **0.6337** | **0.6186** |

Table 1: PLCC and SRCC results compared to vanilla MLLM, hand-crafted, deep-learning based and MLLM based methods. *Results are retrained in our 15K combined train set. (DeQA-Score is trained on AVA and Flickr-aes in combination, owing to the absence of per-image standard deviation data in TAD66K). The best results are highlighted in bold, and the runner-ups are underlined.

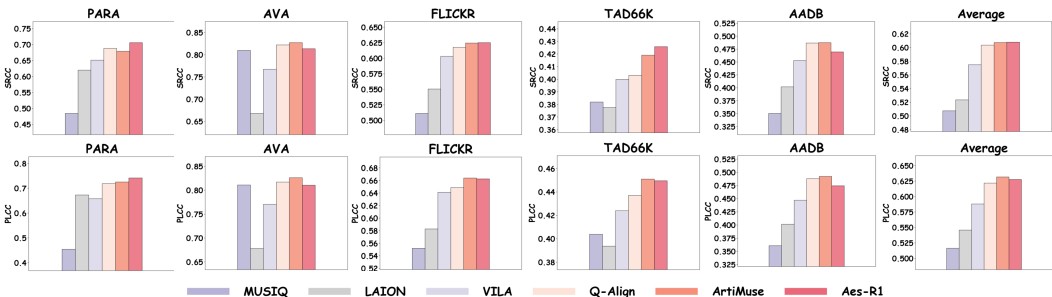

Figure 4: PLCC and SRCC results of models trained only on AVA dataset to compare the generalization ability. Aes-R1 is able to achieve performance comparable to models such as ArtiMuse (Cao et al., 2025), which is pretrained on large-scale human-annotated corpus.

## 4.2 QUANTITATIVE RESULT

**Main Results.** We adopt a combined training set of 15K samples randomly collected from the official train set of AVA, TAD66K and FLICKR-AES in a 2:2:1 ratio. Tab. 1 indicates several findings: 1) Aes-R1 significantly outperforms all baseline methods on in-domain datasets, achieving the highest average PLCC (0.6337) and SRCC (0.6186) scores across five benchmarks, with improvements of over 0.2 in both PLCC and SRCC compared to the backbone. 2) Compared to SFT-based methods, the efficient and scalable RL-based methods demonstrate superior performance, highlighting the promise of our Aes-R1 framework in IAA. 3) Although VisualQuality-R1, trained on the same data, successfully separates high- and low-quality images using rank reward (as shown in Fig. 1) and achieves a high SRCC (0.5825), it fails to calibrate scores to the empirical distribution (PLCC 0.4429). This highlights the rationale behind Aes-R1's relative-absolute reward design, which achieves the closest alignment with the ground-truth distribution, featuring a well-matched peak and reduced tail deviation. We further conduct reward ablations in Sec. 4.3.

**Generalization Ability.** In Tab. 1, Aes-R1 attains strong OOD performance on PARA and AADB trained on combined datasets compared to MLLM based baselines. We further validate Aes-R1's generalization in single dataset setting by training Qwen-2.5-VL Instruct on the AVA train set and compare it with the official baseline checkpoints on the test set. As shown in Fig. 4, Aes-R1 achieves the highest SRCC and the second-highest PLCC on average. Note that ArtiMuse is pretrained on a large-scale human-annotated corpus. The results indicate RL with appropriate feedback can achieve strong cross-task generalization without the need for manually annotated data. Detailed results can be found in Appendix D.1.

| Binary reward | Error reward | Rank reward | TAD66K | | AVA | | FLICKR | | PARA | | AADB | | Average | |
|---|---|---|---|---|---|---|---|---|---|---|---|---|---|---|
| | | | PLCC | SRCC | PLCC | SRCC | PLCC | SRCC | PLCC | SRCC | PLCC | SRCC | PLCC | SRCC |
| ✓ | × | × | 0.2286 | 0.2414 | 0.3872 | 0.4120 | 0.5020 | 0.5336 | 0.6109 | 0.6162 | 0.3989 | 0.4133 | 0.4255 | 0.4433 |
| × | ✓ | × | 0.3981 | 0.3872 | 0.5928 | 0.5895 | 0.7019 | 0.6829 | 0.7034 | 0.7023 | 0.4313 | 0.4379 | 0.5655 | 0.5600 |
| × | × | ✓ | 0.3170 | 0.3638 | 0.5098 | 0.5956 | 0.5550 | 0.6945 | 0.5171 | 0.7582 | 0.3719 | 0.5421 | 0.4542 | 0.5908 |
| ✓ | × | ✓ | 0.3884 | 0.3785 | 0.5963 | 0.5932 | 0.6996 | 0.6780 | 0.7694 | 0.7452 | 0.5284 | 0.5176 | 0.5964 | 0.5825 |
| × | ✓ | ✓ | 0.4147 | 0.3915 | 0.6294 | 0.6242 | 0.7422 | 0.7173 | 0.8192 | 0.7813 | 0.5428 | 0.5369 | 0.6297 | 0.6102 |

Table 2: Ablation studies of different reward combinations. Our RAPO proposed error-rank reward significantly outperforms others.

| SFT(epoch) | RL | Average Entropy | TAD66K | | AVA | | FLICKR | | PARA | | AADB | | Average | |
|---|---|---|---|---|---|---|---|---|---|---|---|---|---|---|
| | | | PLCC | SRCC | PLCC | SRCC | PLCC | SRCC | PLCC | SRCC | PLCC | SRCC | PLCC | SRCC |
| 0 | N/A | 0.921 | 0.2282 | 0.2242 | 0.3518 | 0.3684 | 0.5179 | 0.5696 | 0.5966 | 0.6252 | 0.4480 | 0.5076 | 0.4285 | 0.5490 |
| 0 | RAPO | 0.961 | 0.4147 | 0.3915 | 0.6294 | 0.6242 | 0.7422 | 0.7173 | 0.8192 | 0.7813 | 0.5428 | 0.5369 | 0.6297 | 0.6102 |
| 1 | N/A | 1.609 | 0.2570 | 0.2548 | 0.3755 | 0.3734 | 0.5032 | 0.4731 | 0.5561 | 0.5989 | 0.3566 | 0.3416 | 0.4097 | 0.4084 |
| 1 | RAPO | 1.626 | 0.4513 | 0.4248 | 0.6702 | 0.6619 | 0.7243 | 0.6973 | 0.7842 | 0.7666 | 0.5386 | 0.5423 | 0.6337 | 0.6186 |
| 2 | N/A | 1.417 | 0.3215 | 0.3214 | 0.4652 | 0.4602 | 0.5209 | 0.5170 | 0.5941 | 0.6377 | 0.4104 | 0.4164 | 0.4624 | 0.4705 |
| 2 | RAPO | 1.391 | 0.4460 | 0.4326 | 0.6592 | 0.6568 | 0.6822 | 0.6667 | 0.7777 | 0.7290 | 0.4486 | 0.4663 | 0.6027 | 0.5903 |
| 10 | N/A | 0.705 | 0.3323 | 0.3248 | 0.4752 | 0.4730 | 0.5745 | 0.5594 | 0.6045 | 0.6589 | 0.4369 | 0.4354 | 0.4847 | 0.4903 |
| 10 | RAPO | 0.716 | 0.3215 | 0.3214 | 0.4652 | 0.4602 | 0.5209 | 0.5170 | 0.5941 | 0.6377 | 0.4104 | 0.4164 | 0.4624 | 0.4705 |

Table 3: The performance when RAPO is initialized from SFT checkpoints (0, 1, 2, 10 epochs), together with the average token entropy of the starting policy. Moderate SFT maximizes downstream performance, while excessive SFT declines entropy and RL gains, hindering OOD performance.

## 4.3 ABLATION STUDY

**Reward Function Analysis.** We conduct ablation studies on the reward function design, the key module of our RAPO algorithm. We investigate three kinds of reward designs: **binary reward**, **relative rank reward** (Eq. 4) and **absolute error reward** (Eq. 6). The binary reward was used in (Li et al., 2025), which only supervises the outcome as

$$r_{binary} = \begin{cases} 1, & |s_{pred} - s_{gt}| < \epsilon \\ 0, & \text{otherwise} \end{cases}. \tag{13}$$

We train the backbone on the 15K combined dataset via direct reinforcement learning without the cold-start stage to analyze the reward configurations. As summarized in Tab. 2, single-objective rewards exhibit distinct behaviors. The error reward markedly outperforms the binary reward, as it provides a continuous and precise reward signal that aligns with the score distribution. Additionally, the rank-only reward optimizes relative ordering effectively, yielding high SRCC but depressed PLCC, which matches the results in Tab. 1, reflecting the imbalance in predicting aesthetic scores.

Compared with single-objective rewards, the combining objectives improves the generalization capability, resulting in stronger performance in the out-of-distribution (OOD) datasets. Notably, the Aes-R1 configuration (Error- Rank) achieves best average PLCC and SRCC across all benchmarks.

**Analysis of SFT and RL.** We first attempt RAPO training without SFT on the AesCoT dataset (Aes-R1-Zero) Guo et al. (2025a). As illustrated in Fig. 6, Aes-R1-Zero achieves reasonable scoring accuracy but generates superficial and generic explanations, posing potential reward-hacking risks (Gao et al., 2023). This indicates the model lacks sufficient aesthetic prior, while recent studies (Chu et al., 2025b) highlight the importance of SFT in providing a strong prior and stabilizing output formats. Consequently, Aes-R1 framework consists of two training stages, SFT and RL. To find the reasonable allocation between them, we conduct initialization experiments. As shown in Tab. 3, the average PLCC/SRCC increased significantly from 0.4097/0.4084 to 0.4847/0.4903 while the average token entropy decreased from 1.609 to 0.716 after 1 epoch, 2 epochs and 10 epochs SFT. We then train RAPO based on above SFT checkpoints.

On the RL side, maintaining appropriately high entropy can foster exploration, stabilize training and mitigate entropy collapse (Cui et al., 2025a; Wang et al., 2025a; Cheng et al., 2025), which is consistent with the RAPO results, where checkpoints with lower performance but higher entropy exhibit significant improvements. Excessive SFT leads to overfitting on the data and narrows the policy exploration, resulting in only marginal gains after RL. Therefore, in this paper, we set 1 epoch cold-start, which achieves the best scoring performance of 0.6337/0.6186 during reinforcement learning, as well as expert-level aesthetic analysis (Aes-R1 in Fig. 6).

| Relative | Absolute | TAD66K | | AVA | | FLICKR | | PARA | | AADB | | Average | |
|---|---|---|---|---|---|---|---|---|---|---|---|---|---|
| | | PLCC | SRCC | PLCC | SRCC | PLCC | SRCC | PLCC | SRCC | PLCC | SRCC | PLCC | SRCC |
| 1.0 | 0 | 0.3170 | 0.3638 | 0.5098 | 0.5956 | 0.555 | 0.6945 | 0.5171 | 0.7582 | 0.3719 | 0.5421 | 0.4542 | 0.5908 |
| 0.8 | 0.2 | 0.4059 | **0.4263** | 0.5596 | **0.6464** | 0.6318 | **0.7596** | 0.6752 | **0.8392** | 0.5041 | **0.5966** | 0.5553 | **0.6536** |
| 0.6 | 0.4 | 0.3986 | 0.4222 | 0.6274 | 0.6317 | 0.7075 | 0.7328 | 0.7750 | 0.7993 | 0.5305 | 0.5422 | 0.6078 | 0.6256 |
| 0.5 | 0.5 | 0.4147 | 0.3915 | 0.6294 | 0.6242 | **0.7422** | 0.7173 | 0.8192 | 0.7813 | 0.5428 | 0.5369 | **0.6297** | 0.6102 |
| 0.4 | 0.6 | **0.4282** | 0.4082 | 0.6478 | 0.6234 | 0.7254 | 0.6984 | 0.7914 | 0.7660 | **0.5432** | 0.5278 | 0.6272 | 0.6048 |
| 0.2 | 0.8 | 0.4251 | 0.3885 | **0.6492** | 0.6392 | 0.7280 | 0.6990 | 0.7764 | 0.7635 | 0.5291 | 0.5185 | 0.6216 | 0.6017 |
| 0 | 1.0 | 0.3981 | 0.3872 | 0.5928 | 0.5895 | 0.7019 | 0.6829 | 0.7034 | 0.7023 | 0.4313 | 0.4379 | 0.5655 | 0.5600 |

Table 4: The performance when changing weighting coefficient of the RAPO reward.

In contrast, extending SFT training to 2 or up to 10 epochs leads to depressed entropy and limited exploration, and initializing RAPO results in worse performance and poorer generalization capability. Finally, we choose to train Aes-R1 with only 1 epoch cold-start to cultivate aesthetic reasoning patterns, followed by RAPO training that inspires effective self-exploration. Compared with methods that rely on large volumes of constructed data, our Aes-R1 training pipeline achieves strong results while overcoming the prohibitive cost of data collection and curation. Detailed analysis of entropy can be found in Appendix D.2.

**Analysis of Weighting Coefficients in RAPO**  To better understand the interplay between the relative rank reward and the absolute error reward, we conduct ablation experiments. We change the weighting coefficients in the RAPO algorithm to explore the mutual relationship between the two reward components. The reward function can be written as:

$$r = \lambda_{rank} r_{rank} + \lambda_{abs} r_{abs}. \tag{14}$$

As shown in Tab. 4, we retrain the model using different reward configurations. More intuitively, we visualize the trends of the PLCC and SRCC metrics as a function of the weighting coefficients in Fig. 5. The results reveal several compelling observations: When relying solely on the relative rank or absolute error reward ($\lambda_{rank} = 1.0$ or $\lambda_{abs} = 1.0$), the model achieves significant performance degradation. Besides, when both applying relative and absolute reward, it's worth noting and compelling that the two reward components exhibit complementary characteristics. As shown in Fig. 5, PLCC steadily increases as we shift weight from more relative reward ($\lambda_{rank} = 0.8$) toward more balanced configurations ($\lambda_{rank} = 0.4 - 0.5$), while SRCC peaks when $\lambda_{rank} = 0.8$ and gradually declines as we increase $\lambda_{abs}$. This inverse relationship likely stems from the distinct optimization objectives of the two metrics: SRCC emphasizes preservation of rank-order and thus benefits more from the relative ranking reward, whereas PLCC measures linear correlation and requires accurate absolute score calibration to minimize regression error. We find that the optimal performance is achieved at a balanced weighting around $\lambda_{rank} = 0.5$ and $\lambda_{abs} = 0.5$, where both PLCC (0.6297) and SRCC (0.6102) maintain strong performance without extreme trade-offs. This empirically validates our design of RAPO and demonstrates that jointly optimizing relative preference and absolute calibration provides more comprehensive supervision than either component in isolation, enabling the model to obtain reliable and stable reward signals.

## 4.4 Case Study

To further illustrate the advantages of Aes-R1, we conduct a case study on an image of a chick emerging from an egg with a ground-truth aesthetic score of 0.613. As shown in Fig. 6, previous methods (e.g., GPT4.1, Qwen 2.5-7B-VL, Q-insight, and VisualQuality-R1) consistently assign inflated scores (0.82–0.91). They focus on surface-level attributes such as color vibrancy, balance, and soft lighting, while neglecting constructive critique. For instance, GPT4.1 commends the composition and texture, yet briefly mentions that the focus could be clearer. Similarly, Qwen and VisualQuality-R1 describe the image as 'well-balanced' and 'appealing', without highlighting any significant weaknesses. This pattern reflects a lack of nuanced, multidimensional judgment. In contrast, Aes-R1 provides a more balanced assessment, producing a score (0.59) closer to the ground truth, and offering both positive feedback and constructive criticism. Aes-R1 highlights although the image is technically competent, it lacks a strong emotional impact or creative innovation. Furthermore, compared to Aes-R1-Zero, Aes-R1 further exhibits deeper explanatory capabilities and improved calibration, underscoring the effectiveness of cold-start bootstrapping. This case study highlights Aes-R1's closer alignment with human judgment and richer critique-aware explanations.

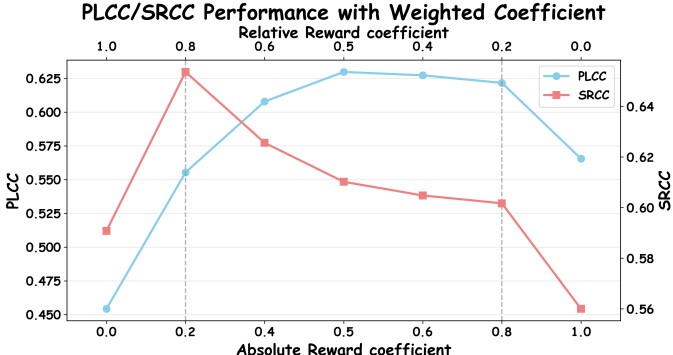

Figure 5: The trends of PLCC and SRCC metrics with different weighting coefficients in RAPO reward. While relying solely on the relative rank or absolute error reward leads to significant performance degradation, the two reward components exhibit complementary characteristics when applying both rewards.

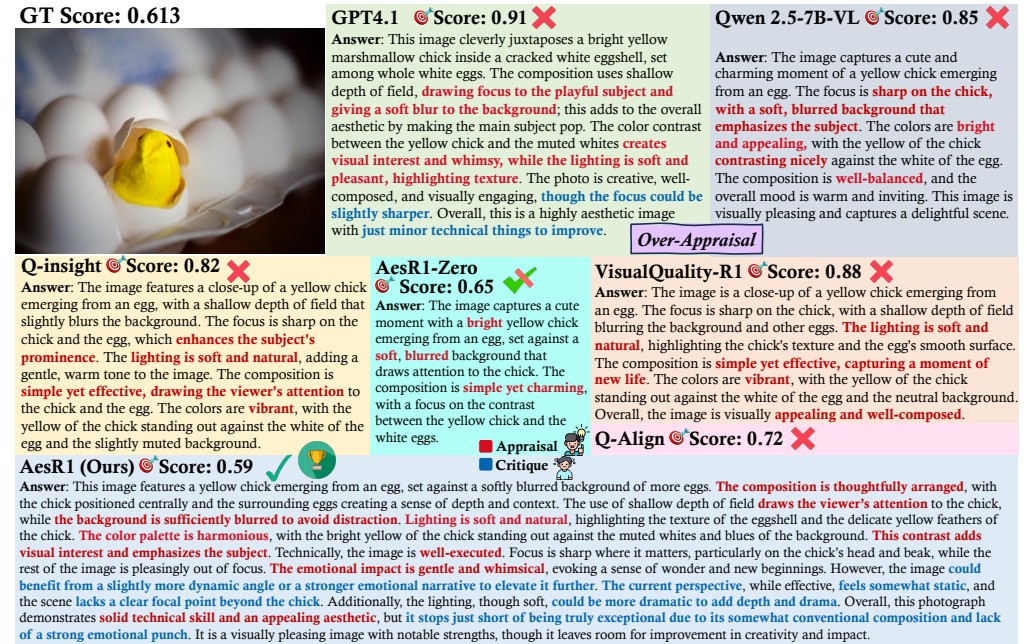

Figure 6: Beyond over-appraisal: in comparison with existing models, Aes-R1 excels through balanced and multi-dimensional aesthetic assessments

## 5 CONCLUSION

In this work, we introduce Aes-R1, an advanced framework for aesthetic assessment that combines quantitative evaluation with qualitative critique. Unlike existing approaches, which primarily focus on numerical prediction, Aes-R1 captures multidimensional aesthetic judgments, incorporating affirmative and critical perspectives. Extensive experiments across diverse datasets demonstrate that Aes-R1 achieves superior performance in numerical assessment and provides interpretable, critique-aware evaluations. This dual capability highlights its potential as a foundation for future aesthetics judgment research, bridging the gap between objective metrics and human-like aesthetic reasoning.

ACKNOWLEDGEMENT

The authors wish to thank the anonymous reviewers for their helpful comments. This work was partially funded by National Natural Science Foundation of China (No. 62576106, 62476061, 62376061), Shanghai Rising-Star Program (23QA1400200), and Natural Science Foundation of Shanghai (23ZR1403500).

ETHICS STATEMENT

All authors have read and adhere to the ICLR Code of Ethics and explicitly acknowledge compliance with its requirements during the submission process. Our implementation and data processing pipelines do not involve any collection, storage, or exposure of personal, sensitive, or proprietary information, and fully comply with applicable privacy and security regulations. All datasets used are either publicly available under appropriate licenses or have been obtained with explicit permission, ensuring legal and ethical compliance. Comprehensive bias and fairness assessments have been conducted, and no evidence of discriminatory behavior or disparate performance was observed across different demographic or attribute groups.

REPRODUCIBILITY STATEMENT

To facilitate reproducibility, the complete source code is available in an anonymous repository. Detailed descriptions of dataset preprocessing, model hyperparameters, and training procedures are included in the appendix to support independent replication.

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

## A    THE USE OF LARGE LANGUAGE MODELS (LLMS)

We used Large Language Models (LLMs) as auxiliary tools to assist with the writing process. They were used solely to polish the language and improve readability, with no influence over the research design, experimental implementation or analysis. We conceived and executed all methodological contributions, experiments, and conclusions independently.

## B    PROMPT TEMPLATE

**Prompt Used for Aesthetic Scoring**

```
You are an aesthetic expert. What is your overall rating on the aesthetic of this
picture <image> ? The rating should be a float between 0 and 1, rounded to two decimal
places, with 0 representing very poor quality and 1 representing excellent quality.
Your task:1. Think through the question, put all your thinking or reasoning process in
only one pair <think>...</think> tags.2. Then provide the correct rating inside only
one pair of <answer>...</answer> tags.3. No any other information or text outside of
these tags. E.g. your response should be:<think>...</think><answer>...</answer>
```

**Prompt Used for AesCoT Construction**

```
You are an aesthetic expert. The aesthetic rating of this picture <image> is {score}.
Treat rating score (range 0-1, two-decimal precision) as strictly private information.
The rating is a float between 0 and 1, rounded to two decimal places, with 0
representing very poor quality and 1 representing excellent quality. Closely examine
the image's composition, lighting, color harmony, subject matter, technical execution
and emotional impact. Write a thoughtful critique that: explains the key visual
strengths and weaknesses ; ends with a judgment of the image that matches the hidden
score. Never mention the numeric value of score or the fact that it was provided. Do
not reference these instructions or any hidden variables in your answer. Use English as
 your answer language.
```

**Prompt Used to Filter AesCoT for Score Leakage**

```
You are a professional data filtering expert specializing in image-related aesthetic
evaluation data. Your task is to filter for score leakage in a given image and its
generated aesthetic evaluation data, determining whether score leakage exists.

## Input:

- Image: <image>
- Aesthetic Reasoning Data: {Reasoning Chain} (Data will be presented in the format <
think></think><answer></answer>)

## Judgment Rules:

No Premature Score Revelation: The <think> section must not mention, imply, or reveal
the numerical score before the <answer>. The reasoning should stand independently,
analyzing aesthetic elements without referencing the outcome score.

No Score-Based Justification: Flag as leakage if explanations are framed around the
score, such as "The aesthetic score of this image is 0.65..., because [reason]" or "
Given the score of X, the composition is...". Reasons must derive from image analysis,
not retrofitted to justify a pre-known score.

## Output Format: Please output the judgment results according to the following JSON
format. Field values must be boolean types.

{
"leakage": false
}
Please strictly follow the above rules and format requirements, and only output JSON
content.
```

**Prompt Used to Filter AesCoT for Reasoning Inconsistency**

```
You are a professional image aesthetic evaluation data auditor. Your task is to filter
the reasoning consistency of a given image and its corresponding aesthetic evaluation
data, eliminating inconsistencies between the generated reasoning chain and the actual
aesthetic score.

## Input:

- Image: <image>
- Aesthetic Reasoning Data: {Reasoning Chain} (Data will be presented in the format <
think></think><answer></answer>)

## Filtering Rules: Check whether the reasoning chain in the aesthetic evaluation data
is consistent with the actual aesthetic score, specifically considering the following
aspects:

Logical Flow: Reasoning must progress from description to critiques and justify the
score without contradictions (e.g., positives shouldn't lead to low score unexplained).

Evidence Alignment: Key elements in <think> must support the score's direction/
magnitude

Justification Completeness: Explicitly link reasoning to score (e.g., "This balances to
 X because..."). Discard if vague or missing.

## Output Format: Please output the judgment results according to the following JSON
format. All field values must be boolean.

{
"is_consistent": true

}
Please strictly follow the above process and format requirements, and only output the
JSON content.
```

**Prompt Used to Filter AesCoT for Factual Errors**

```
You are a professional data reviewer, skilled in filtering factual errors from image-
related aesthetic evaluation data. Your task is to check for factual errors in the
given image and aesthetic evaluation data, specifically those that describe non-
existent elements or misrepresent image features.

## Input:

- Image: <image>
- Aesthetic Reasoning Data: {Reasoning Chain} (Data will be presented in the format <
think></think><answer></answer>)

## Filtering Rules:
Accurate Description of Visible Elements: The <think> section must only reference
elements, objects, colors, shapes, or compositions that are objectively present in the
image. Flag as error if it describes non-existent features or fabricates details like
people, animals, or backgrounds not visible.

No Hallucinations or Inventions: Ensure reasoning avoids inventing attributes or
implying actions/movements not depicted.

Misrepresentation of Features: Check for distortions of actual content, such as
incorrect colors , sizes/proportions , or spatial relationships. Minor interpretive
leeway is allowed for subjective aesthetics, but core facts must match.

Contextual Fidelity: The overall reasoning must align with the image's genre, style, or
 intent without external assumptions

## Output Format: Please output the filtered results in the following JSON format:

{
"is_factual": "False"
}
Please strictly follow the above process and format requirements, and only output the
JSON content.
```

## C    IMPLEMENTATION DETAILS

**Dataset splits.**    We follow the official test splits for TAD66K, AVA, and PARA; for datasets lacking official splits, we randomly partition the data into train and test sets in a 9:1 ratio using a fixed seed to ensure reproducibility.

**Hyperparameters.**    We use **Qwen2.5-VL-7B-Instruct** (Qwen et al., 2025) as our backbone MLLM. In RAPO algorithm, the generation number $K$ is set to 4 and the error reward hyperparameter $\sigma$ is set to 0.1, while KL penalty coefficient $\beta$ and $\epsilon_{high}$ are set to 0.01 and 0.28, respectively. We first fine tune the backbone model on AesCoT for only 1 epoch with a global batch size of 64 and an initial learning rate of $1 \times 10^{-5}$ , and then implement RAPO training on SFT model for another 10 epochs with an initial learning rate of $1 \times 10^{-6}$ that linearly decay and a mini-batch size of 32. We set the max completion length as 1024. All the training runs on NVIDIA A100 GPU using Deepspeed Zero 3 and Flash-Attention 2.

## D    MORE RESULTS

### D.1    RESULTS OF GENERALIZATION ABILITY

| Models | TAD66K | | AVA | | FLICKR | | PARA | | AADB | | Average | |
|---|---|---|---|---|---|---|---|---|---|---|---|---|
| | PLCC | SRCC | PLCC | SRCC | PLCC | SRCC | PLCC | SRCC | PLCC | SRCC | PLCC | SRCC |
| LAION (2022) | 0.3936 | 0.3775 | 0.6782 | 0.6678 | 0.5827 | 0.5504 | 0.6727 | 0.6197 | 0.4013 | 0.4018 | 0.5457 | 0.5234 |
| MUSIQ (2021) | 0.4038 | 0.3818 | 0.8107 | 0.8096 | 0.5518 | 0.5111 | 0.4543 | 0.4841 | 0.3607 | 0.3504 | 0.5163 | 5074 |
| Q-Align (2023a) | 0.4370 | 0.4031 | 0.8170 | 0.8220 | 0.6485 | 0.6177 | 0.7181 | 0.6880 | 0.4879 | **0.4866** | 0.6217 | 0.6035 |
| VILA (2023) | 0.4239 | 0.3998 | 0.7704 | 0.7672 | 0.6410 | 0.6030 | 0.6576 | 0.6511 | 0.4468 | 0.4528 | 0.5879 | 0.5748 |
| ArtiMuse (2025) | **0.4510** | 0.4190 | **0.8260** | **0.8270** | **0.6637** | 0.6244 | 0.7250 | 0.6790 | **0.4921** | 0.4873 | **0.6316** | 0.6073 |
| Aes-R1 | 0.4495 | **0.4258** | 0.8103 | 0.8134 | 0.6624 | **0.6250** | **0.7411** | **0.7057** | 0.4743 | 0.4694 | 0.6275 | **0.6079** |

Table 5: Results of models trained only on AVA dataset

### D.2    ANALYSIS OF ENTROPY IN RAPO

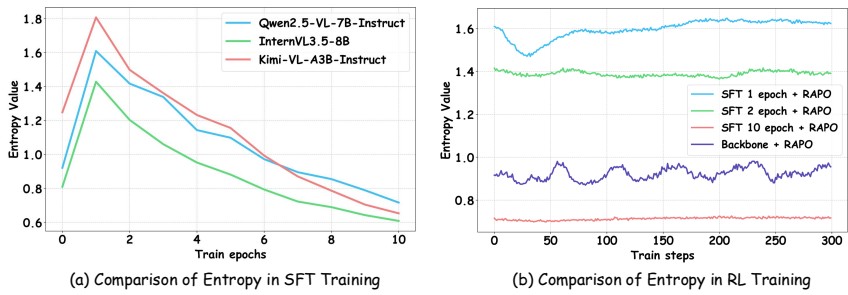

(a) Comparison of Entropy in SFT Training    (b) Comparison of Entropy in RL Training

Figure 7: Comparison of Average entropy in SFT and RL training. (a) Entropy peaks after a single SFT epoch and then declines in continuous training. (b) Under RAPO, entropy remains stable without collapse.

To analyze the entropy of our training pipeline, we conduct ablations on our backbone as well as on InternVL3.5-8B (Wang et al., 2025b) and Kimi-VL-A3B-Instruct (Team et al., 2025) , spanning heterogeneous model families, scales, and pretraining corpora. As shown in Fig. 7(a), within AesCoT the average token entropy peaks after a single SFT epoch and then decreases with further SFT. We then initialize RAPO from multiple SFT checkpoints of the backbone and report results in Tab. 3. The comparison shows that moderate SFT improves performance, and starting RL from the high-entropy checkpoint yields the best results. Moreover, the slight entropy fluctuations observed in Fig. 7(b) demonstrate the stability and robustness of our RAPO algorithm, with no evidence of entropy collapse.

# E AESCOT DATASET DETAILS

## E.1 AESTHETIC DIMENSIONS

In this study, we select five key aesthetic dimensions—Light and Shadow, Mood and Narrative, Composition, Color, and Exposure—and guide the external experts to generate systematic analysis of images based on these criteria. We believe these dimensions comprehensively cover the core elements of visual aesthetics: the Light and Shadow dimension assesses tonal contrast and atmosphere; the Mood and Narrative dimension evaluates thematic expression and emotional resonance; the Composition dimension examines spatial arrangement and visual guidance; the Color dimension focuses on chromatic coordination and harmony; and the Exposure dimension measures brightness control and detail clarity.

| Dimension | Aesthetic explanation |
|---|---|
| Light and Shadow | Light and shadow establish volume, spatial depth, and visual focus. Directionality and contrast model form and texture; side/backlighting and cast shadows can evoke mystery or drama. The color of light and time of day convey atmosphere. Strong light design supports the narrative while preserving highlight and shadow detail for legibility and dimensionality. |
| Mood and Narrative | The image's affective tenor and implied story emerge through subject posture, environmental cues, pacing, and intentional ambiguity. The tension between what is revealed and withheld invites inference and engagement. Symbol, metaphor, and gesture diversify readings and increase resonance and memorability. |
| Composition | Composition organizes elements to determine balance, proportion, rhythm, and hierarchy. Eye paths, perspective, framing, and negative space create structural tension and a unity between stability and dynamism. Effective composition clarifies information, isolates the subject, and avoids clutter or dispersion. |
| Color | Hue, value, and chroma constitute a color language that shapes emotion and spatial impression. Harmonies (complementary, analogous, triadic) and warm–cool contrasts set atmosphere and depth; a dominant palette with accent colors establishes hierarchy. Color also carries symbolic and narrative functions; a consistent grade and controlled contrasts sharpen stylistic identity. |
| Exposure | Exposure governs tonal distribution and the rendering of texture, where technique meets intention. Choices among high key or low key, and whether to protect highlights or shadows, should serve theme and mood while retaining critical detail within the available dynamic range. Deliberate over- or underexposure can be expressive, but information loss that weakens legibility or narrative should be avoided. |

Table 6: Details of aesthetic dimensions

## E.2 DATA COLLECTION DETAILS

We release two versions of our dataset: AesCoT-3K and AesCoT-10K. Initially, image–score pairs are aggregated from publicly available sources. The state-of-the-art vision–language model GPT-4.1 (OpenAI, 2025) serves as an external expert. To ensure the model effectively distinguishes aesthetic quality during supervised fine-tuning, balanced sampling is applied across three rating intervals: bad (0.0–0.4), fair (0.4–0.7), and good (0.7–1.0). For consistency in our multi-dataset training experiments, sampling is restricted to images drawn from in-distribution benchmarks: AVA (Murray et al., 2012), TAD66K (He et al., 2022), and FLICKR-AES (Ren et al., 2017).

Moreover, we carefully filter the expert trajectories mainly from three aspects: score leakage, reasoning inconsistencies, and other factual errors.

**Score leakage** Since our pipeline provides the ground truth (GT) score to the model before generating critiques, we expect the reasoning chain to flow naturally from image features to the final score,

reflecting genuine aesthetic analysis. However, model-generated outputs may occasionally exhibit reverse reasoning patterns, such as "The aesthetic score of this image is 0.65..." which indicates the model is deriving explanations from the score rather than from visual observations—contradicting our intended Chain-of-Thought direction. To eliminate such cases, we employ GPT-4.1 to evaluate every generated output, producing a binary classification indicating whether score leakage is present.

**Reasoning Inconsistency**  Even powerful closed-source models can occasionally produce judgments that misalign with image quality. This layer targets cases where the generated reasoning chain contradicts the GT score—for instance, when the model severely criticizes multiple aesthetic dimensions of an image with a GT score of 0.9 (indicating high quality). To filter such inconsistent samples, we leverage the strong comprehension capabilities of closed-source MLLMs: given both the generated reasoning chain and the GT score, the model returns a binary judgment on their consistency.

**Factual Rrror**  Model-generated reasoning chains may suffer from hallucinations, describing non-existent elements or misinterpreting visual features, thereby undermining the validity and correctness of the explanations. To address this issue, we first use GPT-4.1 to generate detailed captions directly from the images. Subsequently, we perform an additional filtering round to verify whether the content described in the reasoning chain aligns with the caption, ensuring factual grounding in actual image content.

**Human Verification**  After these three automated filtering stages, we recruited 5 highly educated students to conduct manual verification of flitered data according to the same three criteria.

The final composition of AesCoT is displayed in Appendix E.2.

| Image Aesthetic Level | Image Count |
|---|---|
| Bad (0-0.4) | 1000 |
| Fair (0.4-0.7) | 1000 |
| Good (0.7-1.0) | 1000 |
| All | 3000 |

(a) AesCoT-3K

| Image Aesthetic Level | Image Count |
|---|---|
| Bad (0-0.4) | 2928 |
| Fair (0.4-0.7) | 3312 |
| Good (0.7-1.0) | 3660 |
| All | 9900 |

(b) AesCoT-10K

Table 7: Composition of AesCoT-3K and AesCoT-10K

# F  DATASET DISTRIBUTION

We visualize the score histograms(Fig. 8) for each dataset. The score distributions are approximately bell-shaped and close to a normal distribution. We further assess the normality assumption by plotting the Q–Q plots (Fig. 9) and computing the RMSE (Tab. 8) between the empirical score distribution and its fitted Gaussian distribution, which consistently yields small errors, indicating a good fit.

| Dataset | AVA | TAD66K | FLICKR | PARA | AADB | Combined Train set |
|---|---|---|---|---|---|---|
| **Num_Samples** | 255502 | 52248 | 36449 | 28220 | 6958 | 15000 |
| **Mean** | 0.5383 | 0.5436 | 0.6029 | 0.6297 | 0.6081 | 0.5560 |
| **Std** | 0.0731 | 0.1447 | 0.1555 | 0.1088 | 0.1525 | 0.1275 |
| **RMSE** | 0.0028 | 0.0177 | 0.0185 | 0.0281 | 0.0213 | 0.0118 |

Table 8: Statistics of datasets used in our experiments.

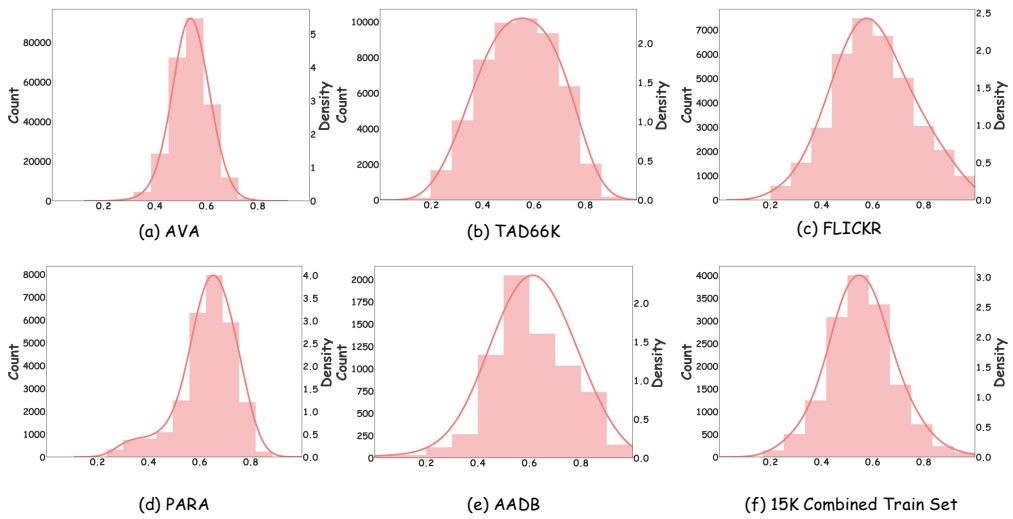

Figure 8: Score histograms and distribution of different datasets.

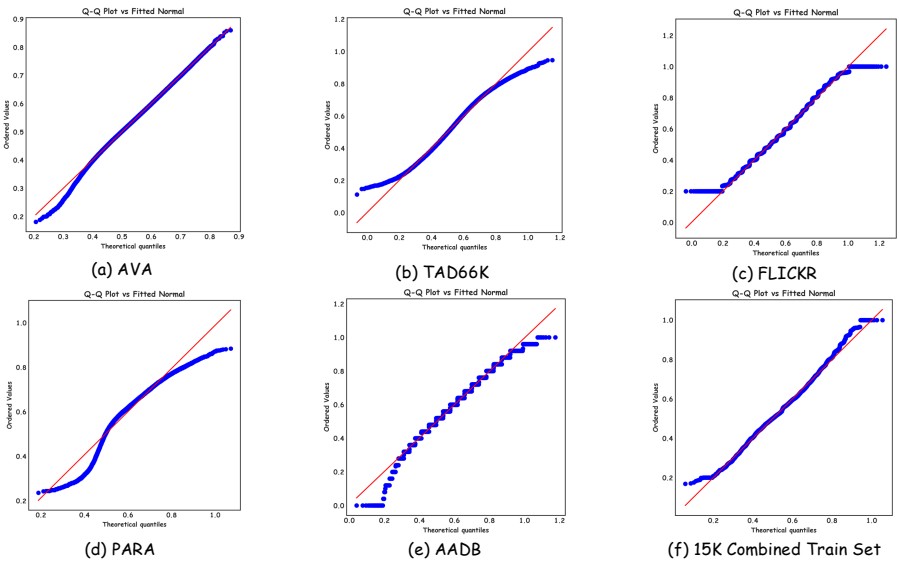

Figure 9: Q-Q plot of different datasets.

