# OpenReview forum: "Unlocking the Essence of Beauty: Advanced Aesthetic Reasoning with Relative-Absolute Policy Optimization"
_ICLR.cc/2026/Conference — ICLR 2026 Poster_

### Official Review · Reviewer_FLJS · 2025-10-27

**Soundness:** 3
**Presentation:** 3
**Contribution:** 2
**Rating:** 6
**Confidence:** 4

**Summary:**

This paper proposes Aes-R1, a novel framework for Image Aesthetic Assessment (IAA) using multimodal large language models (MLLMs). It introduces two key components: (1) AesCoT, an automatic data pipeline that generates high-quality chain-of-thought (CoT) aesthetic critiques along five dimensions (light, composition, color, exposure, mood), and (2) Relative-Absolute Policy Optimization (RAPO), a reinforcement learning (RL) algorithm that jointly optimizes absolute score regression and relative pairwise ranking. The method is trained on only 15K samples and achieves state-of-the-art performance across multiple IAA benchmarks, significantly outperforming existing baselines in both PLCC and SRCC metrics. The paper also provides insightful ablation studies on reward design and the interplay between supervised fine-tuning (SFT) and RL, emphasizing the importance of moderate SFT for maintaining policy entropy and enabling effective RL exploration.

**Strengths:**

* S1: Novel and well-motivated RL framework: RAPO elegantly combines absolute and relative rewards to address the dual nature of human aesthetic judgment—calibrated scoring and ordinal preference—resolving limitations of prior rank-only or score-only RL approaches.

* S2: Aes-R1 achieves significant gains (e.g., +47.9% PLCC) over strong baselines with limited training data and demonstrates robust out-of-distribution generalization.

**Weaknesses:**

* W1: AesCoT relies on GPT-4.1 for data generation, which may limit reproducibility and raise concerns about generalizability if weaker open-source models are used as the "expert."

* W2: It seems that the main contribution of this paper lies in the dataset, and there is not much innovation in terms of methodology.

**Questions:**

NA

---

> ### Author Response · Authors · 2025-11-20
> **Response to Weakness 1 and Weakness 2**
>
> We sincerely for the thoughtful comment and support of our work! Here is the point-by-point response:
>
> **`Weakness 1`** : The problem o f AesCoT Data generation
>
> ---
>
> The annotation of aesthetic data typically relies heavily on expensive expert human annotation which is **difficult to scale**. One of our key contributions is the development of an automated, cost-effective pipeline that leverages existing closed-source models to construct high-quality aesthetic reasoning chains, which already represents a significant reduction in annotation costs compared to traditional expert-driven approaches.
>
> Crucially, our framework treats GPT-4.1 as a replaceable expert teacher model, not as an intrinsic dependency of the methodology itself. The core innovation lies in the AesCoT construction pipeline—including the **structured prompting strategy, multi-dimensional aesthetic dimension design, and three-layer filtering mechanism**—which can be applied with any sufficiently capable vision-language model. We have provided complete prompt templates and filtering strategies (including newly added filtering details in the revised verison pdf in Appendix) to ensure full reproducibility of the data generation process.
>
> Instead of using weaker open-source models, we encourage researchers to adopt more advanced reasoning models than the base model such as GPT-5-Thinking . Our framework's model-agnostic design ensures seamless integration with these emerging capabilities. We will **open-source** both versions of the AesCoT dataset, enabling researchers to reproduce our results and build upon our framework.
>
> **`Weakness 2`** : The innovation and the main contribution
>
> ---
>
> Besides our contribution on dataset, we propose a more general and principled paradigm for preference evaluation, accompanied by a carefully designed data pipeline and reinforcement learning framework.
>
> Traditional approaches in preference learning typically rely on Reward Models (RM) to optimize preference evaluation. However, these methods can only provide **scalar reward signals** and**lack the capability to model the reasoning process**. In contrast, the emergence of Generative Reward Models (GRM) has introduced new possibilities for preference evaluation tasks by generating intermediate reasoning processes and providing richer learning signals for the model. Our method is more closely aligned with the GRM framework, emphasizing the **modeling and optimization of evaluation reasoning processes**. A currently popular pipeline is employing RLVR (Reinforcement Learning via Verifiable Rewards). This paradigm has achieved remarkable success in verifiable tasks such as mathematical reasoning and code generation.
>
> In our pilot study, we discovered that **open-ended evaluation** tasks (such as IAA) fundamentally differ from closed-ended tasks like mathematics and coding. First, open-ended evaluation tasks often lack carefully annotated high-quality reasoning data, making it difficult for models to acquire sufficient domain **prior knowledge** during the SFT phase, consequently hampering their ability to rollout high-quality results worthy of learning from their distribution. Additionally, for open-ended verification tasks, directly using outcome rewards like RLVR often fails to provide stable and reliable reward signals, while process rewards require dense step-level supervision.
>
> Based on these observations, we first designed an automated high-quality reasoning data construction pipeline. Second, to address the pain points of previous Outcome Reward Model (ORM) methods, we propose **Relative-Absolute Policy** **Optimization** **(RAPO)**, specifically designed for **open-ended, highly subjective evaluation tasks**. RAPO provides stable reward signals through relative-absolute joint modeling, without relying on definitive "correct answers," instead learning human evaluation preference patterns.
>
> We selected IAA as our application scenario because it represents an exceptionally challenging open-ended evaluation task requiring **deep reasoning and professional understanding**, making it ideal for demonstrating the superiority of our approach. Our framework **possesses high generalizability** and will be extended to additional open-ended evaluation domains such as creative writing, video quality assessment, and music composition evaluation in future work.
>
> ---
>
> We would like to extend our sincere gratitude for your insightful comments and support! We would be more than willing to address any additional concerns you may have.

---

> ### Author Response · Authors · 2025-11-27
> **Thank you & Looking forward to further discussion**
>
> Dear Reviewer FLJS,
>
> We sincerely appreciate your generous commendation and thorough feedback on our work. To aid in better understanding our rebuttal and revisions, we have summarized your key concerns and our responses below:
>
> **`AesCoT data generation`** We clarify that our AesCoT construction pipeline is an automated, low-cost pipeline that treats GPT‑4.1 as a replaceable expert rather than a hard dependency, with fully presented prompts and filters in our reviesed version  (and planned AesCoT release) to ensure reproducibility and easy substitution with stronger future models like GPT‑5‑Thinking.
>
> **`Innovation & our  main contribution`** We emphasize that, beyond the dataset, our key novelty is a general, principled paradigm specifically designed for open-ended, highly subjective evaluation tasks which is completely distinct from close combining automated reasoning‑chain construction with the RAPO framework. We differentiated open-ended aesthetic tasks from verifiable tasks (like math/code) to justify why our RAPO framework is uniquely necessary and superior to standard RLVR pipelines in this context.
>
> For any other issues not mentioned here, please refer to our detailed rebuttal response. We sincerely and humbly believe we have responded to your concerns properly, and we once again thank you for your meticulous and generous review!
>
> Warm regards,
>
> Authors

---

### Official Review · Reviewer_xmgr · 2025-10-31

**Soundness:** 2
**Presentation:** 2
**Contribution:** 3
**Rating:** 4
**Confidence:** 3

**Summary:**

This paper introduces Aes-R1, a two-stage framework for advanced Image Aesthetic Assessment (IAA). It first "cold-starts" an MLLM by fine-tuning it on AesCoT, a new dataset of high-quality aesthetic critiques, to instill reasoning patterns. It then applies a novel reinforcement learning algorithm, Relative-Absolute Policy Optimization (RAPO), which uses a unique reward function that jointly optimizes for both absolute score accuracy and relative ranking preference. This combined approach allows Aes-R1 to significantly outperform existing baselines, improving its backbone model's average PLCC/SRCC scores by 47.9%/34.8% with minimal training data.

**Strengths:**

1. Novel Relative-Absolute Policy Optimization (RAPO) Algorithm
The paper introduces RAPO, a novel RL algorithm with a joint reward function that optimizes for both "absolute scores" and "relative ranking." This design better reflects complex human aesthetic judgment.

2. Effective Two-Stage "SFT Cold-Start + RL Finetuning" Framework
The authors designed a two-stage process that first uses SFT to inject necessary aesthetic reasoning knowledge, then uses RL to align the model's scoring preferences.

**Weaknesses:**

1. **Reasoning Quality Is Not Quantitatively Evaluated.**
The paper's central claim—that SFT+RL training produces superior aesthetic reasoning compared to pure RL—is supported only by a single, qualitative case study. The work lacks any systematic, quantitative evaluation of the generated "aesthetic analysis text." All metrics (PLCC/SRCC) focus exclusively on the final score's accuracy, failing to validate the "advanced reasoning" contribution mentioned in the title.
2. The AesCoT reasoning data is generated by GPT-4.1, which the authors treat as a competent external source. However, GPT-4.1 performs poorly on direct aesthetic score prediction (as shown in Table 1), raising questions about its suitability as a source of high-quality aesthetic critique. While the paper does include a filtering process, a more thorough validation of AesCoT's explanation quality—either through human evaluation or comparison to expert-written analyses—would help justify the choice of GPT-4.1 as a reasoning teacher.
3. The RAPO framework combines the relative and absolute rewards using a simple summation ($r = r_{rank} + r_{abs}$), which implies an equal 1:1 weighting. While this combined reward proves effective, the paper does not explore the relative influence of these two components. It would be valuable to include an ablation study on the weighting coefficients (e.g., $\alpha \cdot r_{rank} + \beta \cdot r_{abs}$) to better understand their interplay and to investigate whether a different balance could further optimize performance.
4. The SFT cold-start phase is presented as critical, but its implementation lacks clarity. The authors mention both AesCoT-3K and AesCoT-10K datasets in the appendix but fail to specify which version was actually used for the 1-epoch SFT in the main experiments. This ambiguity makes it impossible to reproduce the key results or understand the precise data conditions for the model's "cold start".

**Questions:**

1. Regarding the SFT cold-start phase: The appendix mentions both AesCoT-3K and AesCoT-10K, but the main paper does not specify which was used for the 1-epoch SFT. To ensure the reproducibility of the key results, could the authors please clarify which dataset version (3K or 10K) was used?

**Suggestion:**
There might be a typo on Line 142 ("fro" -> "for").

---

> ### Author Response · Authors · 2025-11-20
> **Response to Weakness 1**
>
> Thank you for your thorough review and insightful comments on our manuscript! We have carefully considered your suggestions and have made the following revisions and clarifications to address the concerns raised.
>
> **`Weakness 1`** : Reasoning quality is not quantitatively evaluated.
>
> ---
>
> We sincerely appreciate your valuable feedback on the need for quantitative evaluation of reasoning quality. In response to this concern, we have made every effort within our available resources and time constraints to conduct both human and model-based evaluations to rigorously assess the reasoning quality of our framework.
>
> 1. **Human Annotation**
>
> Due to limited time and budget, we urgently recruited 5 graduate students with advanced education backgrounds to annotate reasoning quality on a sample of 100 instances uniformly drawn from the test sets of five benchmark datasets. For each sampled image, we generated reasoning outputs using five different models (including our Aes-R1 and baseline methods). We then presented these outputs to the annotators in a blind, multiple-choice format and computed the human agreement rate for each model, measuring which reasoning chain annotators found most aligned with their own aesthetic judgments. This human evaluation directly reflects the alignment between model reasoning and expert human perception
>
> |Models|Human agreement rate|
> |---|:---:|
> |Qwen2.5-7b-Instruct|12%|
> |Qwen2.5-32b-Instruct|21%|
> |VisualQuality-R1|17%|
> |Q-Insight|16%|
> |Aes-R1|**34%**|
>
> 2. **Model-Based Evaluation**
>
>
> In addition, to scale our evaluation and provide more granular insights, we employed GPT-5 as an automated evaluator to assess reasoning quality on a larger sample of 500 instances.
>
> We designed a comprehensive evaluation rubric spanning 8 dimensions across 3 categories
>
> - **Aesthetic Professionalism** (aesthetic attribute coverage, evaluation accuracy, use of professional terminology)
>
> - **Textual Expression Quality** (fluency, informativeness, consistency)
>
> - **Additional Functional Aspects** (quality of improvement suggestions, emotional exploration)
>
>
> Each dimension was scored from 1–5. We computed the average scores across all dimensions for each model and visualized the results using radar charts in Appendix D.2 to provide an intuitive comparison of strengths and weaknesses across different reasoning aspects.
>
>
>
> ||AAC|EA|PTU|F|IR|IC|IS|EE|Average|
> |---|---|---|---|---|---|---|---|---|---|
> |Qwen2.5-7b-Instruct|3.23|4.04|2.35|4.83|3.09|4.99|1.42|2.73|3.34|
> |Qwen2.5-32b-Instruct|3.87|4.71|3.08|4.92|3.94|4.99|1.91|2.96|3.79|
> |VisualQuality-R1|3.93|4.77|2.79|4.85|3.68|4.98|1.58|3.22|3.73|
> |Q-Insight|3.69|4.73|2.77|4.84|3.62|4.99|1.30|3.19|3.64|
> |Aes-R1|**4.42**|**4.86**|**3.72**|**4.94**|**4.04**|**4.99**|**3.14**|**3.69**|**4.23**|
>
> **AAC**: Aesthetic Attribute Coverage  **EA**: Evaluation Accuracy   **PTU**: Professional Terminology Usage   **F**: Fluency  **IR**: Information Richness **IC**: Internal Consistency **IS**: Improvement Suggestions **EE**: Emotional Exploration
>
> As shown in the tables above and Fig. 8 in Appendix, our Aes-R1 model achieves highest human agreement rate and is consistently best under multidimensional model evaluation.

---

> ### Author Response · Authors · 2025-11-20
> **Response to Weakness 2**
>
> **`Weakness 2`** The suitability of using GPT4.1 as a source of high-quality aesthetic critique
>
> ---
>
> OpenAI reports that GPT-4.1 represents a significant advancement in visual intelligence compared to GPT-4o [1], validating its suitability as a data source. However, in our pilot experiments, we discover that even powerful closed-source models, while excellent at recognizing visual content and understanding aesthetic elements, often struggle to achieve high scoring accuracy in zero-shot. The AesCoT construction process strategically leverages GPT-4.1's strengths while circumventing its weaknesses: we exploit its **robust aesthetic understanding, multi-dimensional reasoning capabilities, structured planning abilities, and superior text generation quality** to construct detailed reasoning chains, while deliberately avoiding direct reliance on its numerical scoring outputs. After applying our three-stage filtering pipeline (score leakage detection, reasoning inconsistency filtering, and factual error elimination), all retained data consist of high-quality, multi-dimensional reasoning chains that are consistent with ground-truth scores. We provide the prompts and details of our filtering process in our revised version pdf.
>
> In addition, the expert-annotated aesthetic critiques would be costly and hard to scale, which demonstrate the advantage of our automatic generation and filtering pipeline. Our choice of GPT-4.1 balances performance and cost. Importantly, in practical application, our framework is model-agnostic: as even more capable models (e.g., GPT5-thinking) become available, they can be seamlessly integrated into our pipeline to further enhance data quality without architectural modifications. To validate the suitability and the quality of our choice, we conducted comparative analyses across multiple data sources. Specifically, we sampled 100 instances each from reasoning chains generated by GPT-4.1, Seed1.5 thinking VL, and Qwen2.5-VL-32B, then we presented these reasoning chains to the 5 well-educated human annotators in a blind, multiple-choice format and computed the human agreement rate to measure the data quality.
>
> ||Human agreement rate|
> |---|---|
> |Seed1.5-thinking-VL|41%|
> |Qwen2.5-VL-32B|6%|
> |GPT 4.1-2025-04-14|**53%**|
>
> Additionally, to evaluate we trained models using cold start data generated by GPT-4.1 and Seed1.5-thinking-VL separately, and measured their downstream aesthetic scoring accuracy. We are pleased to share you with the performance of the models using different data sources after 1 epoch cold start and after both cold start and RAPO training.
>
> After 1 epoch cold start:
>
> | | | | | | ||| |   |   |  |   |   |
> | ----| -----| ---- | ----| ---- | --- | ---| ---| --- | ---| ---| ---| --- | ---|
> |    | **Entropy** | TAD66K   | | AVA      |  | FLICKR   || PARA     |  | AADB     |          | Average  |          |
> |  | | **PLCC** | **SRCC** | **PLCC** | **SRCC** | **PLCC** | **SRCC** | **PLCC** | **SRCC** | **PLCC** | **SRCC** | **PLCC** | **SRCC** |
> | Seed-1.5-thinking-VL | 0.904       | 0.2406   | 0.2293   | **0.4075**   | **0.3986**   | 0.4864   | 0.4717   | 0.4950   | 0.5492   | 0.3281   | 0.3349   | 0.3915   | 0.3967   |
> | GPT4.1   |**1.609**       | **0.2570**   | **0.2548**   | 0.3755   | 0.3734   | **0.5032**   | **0.4731**   | **0.5561**   | **0.5989**   | **0.3566**   | **0.3416**   | **0.4097**   | **0.4084**   |
>
> After cold start +RAPO training :
>
> | | | | | | ||| |   |   |  |   |   |
> | ----| -----| ---- | ----| ---- | --- | ---| ---| --- | ---| ---| ---| --- | ---|
> |                      | TAD66K     |            | AVA        |            | FLICKR     |            | PARA       |            | AADB       |            | Average    |            |
> |                      | **PLCC**   | **SRCC**   | **PLCC**   | **SRCC**   | **PLCC**   | **SRCC**   | **PLCC**   | **SRCC**   | **PLCC**   | **SRCC**   | **PLCC**   | **SRCC**   |
> | Seed-1.5-thinking-VL | 0.3556     | 0.3197     | 0.5526     | 0.5488     | 0.6309     | 0.6118     | 0.7029     | 0.6914     | 0.4078     | 0.4498     | 0.5300     | 0.5243     |
> | GPT4.1               | **0.4513** | **0.4248** | **0.6702** | **0.6619** | **0.7243** | **0.6973** | **0.7842** | **0.7666** | **0.5386** | **0.5423** | **0.6337** | **0.6186** |
>
>
> These experimental results demonstrate that GPT-4.1 serves as a reliable cold-start data source. Compared to other models, when initialized with data generated through our AesCoT pipeline, the model not only exhibits superior initial performance but also maintains higher token entropy, consistent with our analysis in Section 4.3. This elevated entropy facilitates better exploration during the RL phase and enables the model to achieve a higher performance ceiling.
>
> [1] Assessing the Aesthetic Evaluation Capabilities of GPT-4 with Vision: Insights from Group and Individual Assessments
>
> [2] https://openai.com/index/gpt-4-1/

---

> ### Author Response · Authors · 2025-11-20
> **Response to Weakness 3 and Weakness 4**
>
> **`Weakness 3`** **： An ablation study on the weighting coefficients**
>
> ---
>
> Your idea is very constructive! We follow your suggestion and conduct an ablation study.
>
> We change the weighting coefficients in the RAPO algorithm to explore the interplay and mutual relationship between the two reward components (relative ranking and absolute regression). The experimental results are as follows:
>
>
> | |  |  |  |   |   |   |    |   |  |   |  |   |   |
> | --- | ---| ---| ---| --- | ---| ---| ---| --- | ---| --- | ---| ---| ---|
> | Relative | Absolute | TAD66K     | | AVA   |  | FLICKR     |  | PARA   |   | AADB   |  | Average    |            |
> |    |    | **PLCC**   | **SRCC**   | **PLCC**   | **SRCC**   | **PLCC**   | **SRCC**   | **PLCC**   | **SRCC**   | **PLCC**   | **SRCC**   | **PLCC**   | **SRCC**   |
> | 1        | 0        | 0.317      | 0.3638     | 0.5098     | 0.5956     | 0.555      | 0.6945     | 0.5171     | 0.7582     | 0.3719     | 0.5421     | 0.4542     | 0.5908     |
> | 0.8      | 0.2      | 0.4059     | **0.4263** | 0.5596     | **0.6464** | 0.6318     | **0.7596** | 0.6752     | **0.8392** | 0.5041     | **0.5966** | 0.5553     | **0.6536** |
> | 0.6      | 0.4      | 0.3986     | 0.4222     | 0.6274     | 0.6317     | 0.7075     | 0.7328     | 0.7750     | 0.7993     | 0.5305     | 0.5422     | 0.6078     | 0.6256     |
> | 0.5      | 0.5      | 0.4147     | 0.3915     | 0.6294     | 0.6242     | **0.7422** | 0.7173     | **0.8192** | 0.7813     | 0.5428     | 0.5369     | **0.6297** | 0.6102     |
> | 0.4      | 0.6      | **0.4282** | 0.4082     | 0.6478     | 0.6234     | 0.7254     | 0.6984     | 0.7914     | 0.7660     | **0.5432** | 0.5278     | 0.6272     | 0.6048     |
> | 0.2      | 0.8      | 0.4251     | 0.3885     | **0.6492** | 0.6392     | 0.7280     | 0.6990     | 0.7764     | 0.7635     | 0.5291     | 0.5185     | 0.6216     | 0.6017     |
> | 0        | 1        | 0.3981     | 0.3872     | 0.5928     | 0.5895     | 0.7019     | 0.6829     | 0.7034     | 0.7023     | 0.4313     | 0.4379     | 0.5655     | 0.5600     |
>
> More intuitively, we visualize the trends of PLCC and SRCC metrics as a function of the weighting coefficients in the Section 4.3 Figure 5. The results reveal several compelling observations: When relying solely on the relative rank or absolute error reward ($\lambda_{rel} = 1.0$ or $\lambda_{abs}=1.0$), the model achieves **significant performance degradation**. Besides, when both applying relative and absolute reward,it's worth noting and compelling that the two reward components exhibit **complementary characteristics**. As shown in the figure, PLCC steadily increases as we shift weight from more relative reward ($\lambda_{rel}=0.8$) toward a more balanced configurations ($\lambda_{rel}=0.4–0.5$), while SRCC peaks  when $\lambda_{rel}=0.8$ and gradually declines as we increase $\lambda_{abs}$. This inverse relationship likely stems from the distinct optimization objectives of the two metrics: SRCC emphasizes rank-order preservation and thus benefits more from the relative ranking reward, whereas PLCC measures linear correlation and requires accurate absolute score calibration to minimize regression error.
>
> We find that the optimal performance is achieved at a balanced weighting around $\lambda_{rel}=0.5$ and $\lambda_{abs}=0.5$, where both PLCC (0.6297) and SRCC (0.6102) maintain strong performance without extreme trade-offs. This empirically validates our design of RAPO and demonstrates that jointly optimizing relative preference and absolute calibration provides more **comprehensive supervision** than either component in isolation, enabling the model to obtain **reliable and stable** reward signals.
>
> **`Weakness 4`** The clarity of SFT cold-start phase
>
> ---
>
> Thank you for pointing out the lack of clarity and helping to improve our paper!
>
> To clarify, we use **AesCoT-3K** as the SFT training data in our cold-start process. During this phase, we carefully analyzed the impact of different training intensities on model entropy to prevent over-fitting while ensuring the model acquires foundational reasoning capabilities. Through systematic ablation studies in Section 4.3, we determined that a 1-epoch training strategy strikes the optimal balance: it equips the model with basic aesthetic reasoning abilities while maintaining sufficiently high entropy to facilitate effective exploration during the subsequent RL phase. We will **open-source** both versions of the AesCoT dataset, enabling researchers to reproduce our results and build upon our framework.
>
> **`Suggestion`**:
> Thank you for kindly pointing out our typo! We have already corrected it in our revised version.
>
> ---
>
> Finally, we would like to express our sincere gratitude for your active engagement and instructive feedback ! We have dedicated our utmost effort and resources to diligently address each of your concerns. We greatly appreciate this opportunity for revision and highly value your insightful feedback.

---

> ### Author Response · Authors · 2025-11-27
> **Thank you & Looking forward to further discussion**
>
> Dear Reviewer xmgr,
>
> We deeply appreciate your dedication to engaging in author-reviewer discussions. To aid in better understanding our rebuttal and revisions, we have summarized your key concerns and our responses below:
>
> **`No quantitative reasoning evaluation`** We have made every effort within our available resources and time constraints to conduct both human and model-based evaluation, showing that Aes‑R1’s reasoning chains achieve the highest alignment with expert aesthetic judgments across multiple dimensions.
>
> **`Suitability of GPT‑4.1 as critique source`** We justified using GPT‑4.1 via a filtered AesCoT pipeline, human agreement comparisons against alternative models, and we conduct ablation experiemnt in which downstream performance/entropy analyses demonstrates it is a reliable and scalable cold‑start data source.
>
> **`Ablation on weighting coefficients`** We  followed your suggestions and diligently conducted an ablation over RAPO’s relative/absolute weights and showed that purely relative or purely absolute rewards hurt performance, while a balanced setting yields the best PLCC/SRCC. Our visualization display a surpriseing trend confirming the complementarity between absolute and relative reward, which further indicates the superiority of our method.
>
> **`Clarity of SFT cold‑start phase`**  We clarified that we use AesCoT‑3K for a 1‑epoch SFT cold start, justified this choice via entropy and performance analysis, and we will open‑source both AesCoT versions for reproducibility.
>
> For any other issues not mentioned here, please refer to our detailed rebuttal response. We sincerely and humbly believe we have responded to your concerns properly, and we once again thank you for your meticulous and generous review!
>
> Warm regards,
>
> Authors

---

### Official Review · Reviewer_XmWM · 2025-10-31

**Soundness:** 2
**Presentation:** 3
**Contribution:** 2
**Rating:** 4
**Confidence:** 3

**Summary:**

Authors proposed Aes-R1. It is a comprehensive framework for Image Aesthetic Assessment that combines an aesthetic reasoning data pipeline (AesCoT) with a reinforcement learning algorithm (RAPO). It jointly optimizes absolute aesthetic scores and relative ranking consistency, significantly improving both accuracy and generalization.

**Strengths:**

1. The authors build AesCoT, an automatic pipeline that generates high-quality aesthetic reasoning data efficiently.
2. They propose Aes-R1, a two-stage framework using RAPO to jointly optimize absolute scoring and relative ranking.
3. Extensive experiments show Aes-R1 achieves state-of-the-art performance and generalization with limited training data.

**Weaknesses:**

1. The authors modify the reward design based on GRPO to fit the specific characteristics of the IAA task. However, since Q-Insight has adopted similar methods on the same task, the contribution and novelty of this paper appear somewhat diminished.

2. Compared with models that directly predict scores, this reasoning-based MLLM approach requires significantly more inference time, sometimes tens of times longer or even more, which poses clear challenges for large-scale inference. How do the authors consider this inefficiency issue?

3. The presentation in Figure 5 is somewhat unclear. Given that score distributions can vary greatly across datasets, is it reasonable or meaningful to compare their absolute values directly?

**Questions:**

1. Compared with models that directly predict scores, this reasoning-based MLLM approach requires significantly more inference tim, sometimes tens of times longer or even more—which poses clear challenges for large-scale inference. How do the authors consider this inefficiency issue?

2. The presentation in Figure 5 is somewhat unclear. Given that score distributions can vary greatly across datasets, is it reasonable or meaningful to compare their absolute values directly?

---

> ### Author Response · Authors · 2025-11-20
> **Response to Weakness 1**
>
> Thank you for your insightful comments and suggestions! Here is the point-by-point response:
>
> **`Weakness 1`** : The contribution and novelty of our proposed framework.
>
> ---
> 1. **What is the difference between our framework and Q-insight?**
>
> 1.1 **Limitations of Q-insight**
>
> Q-Insight[1] employs a standard RLVR pipeline to guide the model in assessing image quality, using GRPO with an outcome-based reward. However, this design has two key limitations in highly subjective image assessment setting: (i) for the reasoning process, it directly applies RL without a cold-start stage to instill structured reasoning patterns and **prior aesthetic knowledge**, making it difficult for the base model to acquire a robust and expert-level aesthetic reasoning paradigm; and (ii) it implicitly treats image assessment in the same way as closed-form tasks such as mathematics or coding, whereas image quality and aesthetic assessment are inherently **open-ended and highly subjective**, so an ORM-based binary outcome reward alone struggles to provide stable, reliable, and cognitively plausible supervision signals, and fails to learn a **well-calibrated predictive distribution** over aesthetic scores (Figure 1).
>
> 1.2 **Our method**
>
> To address these issues, we first introduce a cold-start stage based on automatically generated high-quality aesthetic reasoning chains, which equips the model with explicit aesthetic priors and structured reasoning abilities before reinforcement learning. Second, to tackle the limitations of prior Outcome Reward Model (ORM) approaches, we propose **Relative-Absolute Policy Optimization (RAPO)**, which jointly optimizes relative ranking and absolute score calibration and is specifically tailored to **open-ended, highly subjective evaluation tasks**. As shown in the score distribution comparison in **Fig. 1**, Aes-R1 produces predicted score distributions that more closely align with the empirical ground-truth than Q-Insight and other image assessment baselines. In addition, the reward-function ablation in **Table 2** demonstrates that a purely binary outcome reward is insufficient to guide the model in open-ended tasks such as aesthetic reasoning, whereas our combined relative–absolute reward yields more stable learning signals and substantial performance gains. Finally, we would like to present our experiment results compared with the method of Q-Insight, with RL training implemented without and after AesCoT cold start. With in both conditions, our proposed Aes-R1 significantly outperforms Q-Insight across five benchmarks.
>
> RL with cold start:
>
> ||||||||||||||
> |-|-|-|-|-|-|-|-|-|-|-|-|-|
> ||**TAD66K**||**AVA**||**FLICKR**||**PARA**||**AADB**||**Average**||
> ||PLCC|SRCC|PLCC|SRCC|PLCC|SRCC|PLCC|SRCC|PLCC|SRCC|PLCC|SRCC|
> |Q-Insight|0.2286|0.2414|0.3872|0.4120|0.5020|0.5336|0.6109|0.6162|0.3989|0.4133|0.4255|0.4433|
> |Aes-R1|**0.4147**|**0.3915**|**0.6294**|**0.6242**|**0.7422**|**0.7173**|**0.8192**|**0.7813**|**0.5428**|**0.5369**| **0.6297**|**0.6102**|
>
> RL After cold start:
>
> ||||||||||||||
> |-|-|-|-|-|-|-|-|-|-|-|-|-|
> ||**TAD66K**||**AVA**||**FLICKR**||**PARA**|| **AADB**||**Average**||
> ||PLCC|SRCC|PLCC|SRCC|PLCC|SRCC|PLCC|SRCC|PLCC|SRCC|PLCC|SRCC|
> |Q-Insight|0.3556|0.3197|0.5398 |0.5453|0.6876|0.6612 |0.7745|0.7428|0.5069|0.5184| 0.5954|0.5813|
> |Aes-R1|**0.4513**|**0.4248**|**0.6702**|**0.6619**|**0.7243**|**0.6973**|**0.7842**|**0.7666**|**0.5386**|**0.5423**| **0.6337**|**0.6186**|
>
> [1] Q-Insight: Understanding Image Quality via Visual Reinforcement Learning
>
> 2. **What is our main contribution and novelty？**
>
> Our main contribution and novelty lie in proposing a more general and principled paradigm for preference evaluation in **open-domain** settings. Traditional Reward Models (RM) can only provide scalar reward signals and lack the capability to model the reasoning process. In contrast, Generative Reward Models (GRM) has introduced new possibilities for preference evaluation tasks by generating intermediate reasoning processes and providing richer learning signals for the model. Our method is more closely aligned with the GRM framework, emphasizing the modeling and optimization of evaluation reasoning processes
>
> While mainstream RLVR pipelines with outcome-only rewards often fail to provide reliable supervision signals, we design a tailored data pipeline and reinforcement learning framework that, first, explores automated, low-cost synthesis of reasoning-chain data in data-sparse regimes, and then applies a relative–absolute joint optimization strategy to make the model’s evaluations better aligned with human perceptual preferences. We validate this general paradigm in the challenging domain of Image Aesthetic Assessment (IAA), which requires **deep understanding and aesthetic reasoning**, representing (to our knowledge) the first application of such a general preference-evaluation scheme to IAA, and we further highlight its potential extension to other open-ended tasks such as creative writing.

---

> ### Author Response · Authors · 2025-11-20
> **Response to Weakness 2 and Weakness 3**
>
> **`Weakness 2`** ：The inference inefficiency issue for reasoning-based MLLM.
>
> ---
>
> Thank you for this thoughtful concern!
>
> 1. **Why reasoning is worth**
>
> For MLLMs, a score-only prediction paradigm is inherently lacking in interpretability: both in practical deployment and during model development, such models behave as black boxes, offering little insight into why a particular aesthetic score is assigned. In contrast, a reasoning-chain-based approach not only contributes to more reliable final scoring, but also provides transparency and interpretability, leading to higher alignment with human aesthetic judgments. Moreover, explicit aesthetic reasoning enables broader downstream applications, such as explainable aesthetic recommendation and human-in-the-loop image creation or editing assistance. In addition, a reasoning-based model can itself serve as a generator of high-quality synthetic training data, forming a virtuous cycle in which better reasoning improves data quality, which in turn further enhances the model.
>
> 2. **How to handle the inference inefficiency issue**
>
> The latency and computational overhead of reasoning models is a common challenge for all large reasoning models (LRMs), and a complete solution is beyond the scope of this work. Fortunately, there is already a growing body of work on test-time and chain-of-thought compression [1][2][3]. In practical production scenarios, we can flexibly combine our framework with these existing compression techniques to dynamically control the length and granularity of the reasoning chains, thereby trading off latency and cost against interpretability and performance according to specific application requirements.
>
> [1] C3oT: Generating Shorter Chain-of-Thought without Compromising Effectiveness
>
> [2]TokenSkip: Controllable Chain-of-Thought Compression in LLMs
>
> [3] LightThinker: Thinking Step-by-Step Compression
>
> **`Weakness 3`**：The clarification of Figure 5 (now Figure 6)
>
> ---
>
> Thank you for pointing out the potential ambiguity in Figure 5 (now Figure 6)！
>
> First, we would like to clarify that Figure 5 (now Figure 6) presents a **case study** on a single image sample. In this example, we compare the reasoning process and the absolute aesthetic scores produced by different baselines. Our goal is to illustrate, in a interpretable way, how the proposed Aes-R1 framework enhances aesthetic reasoning: it can analyze multiple aesthetic dimensions, identify concrete shortcomings of the image, and provide a more comprehensive and structured reasoning chain compared with prior methods.
>
> Second, we agree that different datasets may involve distinct image domains and scoring protocols, which can lead to variations in annotation criteria and score distributions. This domain discrepancy is precisely a long-standing challenge for SFT-based image assessment methods: they tend to overfit a single dataset and struggle to generalize to other benchmarks. Moreover, when using RL, treating image assessment purely as an outcome-supervised, pointwise regression problem encourages the model to over-rely on scalar scores, which does not align well with how humans perceive and compare aesthetic quality. In contrast, our RAPO algorithm jointly captures both the relative preference structure and the absolute score calibration of human judgments, enabling the model to perform strongly on in-domain datasets while also achieving substantial performance gains on out-of-distribution benchmarks **(Table 1)**, which are valued by PLCC/SRCC respectively rather than absolute score.
>
> ---
>
> Thank you very much for your time and efforts in the engagement and insightful comments! We sincerely hope this addresses your concerns and we respectfully look forward to further discussion with you.

---

> ### Author Response · Authors · 2025-11-27
> **Thank you & Looking forward to further discussion**
>
> Dear Reviewer XmWM,
>
> We would like to extend heartfelt thanks to you for your time and efforts in the engagement of the author-reviewer discussion. To facilitate a better understanding of our rebuttal, we hereby summarize your key concerns and our responses as follows:
>
> **`Difference vs. Q-Insight & Contribution`** ：We highlight the limitations of Q-Insight method and position our framework as a more general and principled paradigm for preference evaluation in open-domain settings with superior performance and advantages. We also present the experiment results supporting our conclusions.
>
> **`The inference inefficiency issue for reasoning-based MLLM`**  We argue that explicit reasoning brings interpretability and better alignment, and can be combined with existing CoT-compression methods to flexibly control inference cost.
>
> **`The clarification of Figure 5 (now Figure 6)`** We clarify and explain the presentation in the figure as a single‑image case study, and we further disscussed the advantages of our method when score distributions vary greatly across datasets.
>
> For any other issues not mentioned here, please refer to our detailed rebuttal response. We sincerely and humbly believe we have responded to your concerns properly, and we once again thank you for your meticulous and generous review!
>
> Warm regards,
>
> Authors

---

### Official Review · Reviewer_zj9p · 2025-11-01

**Soundness:** 3
**Presentation:** 3
**Contribution:** 2
**Rating:** 4
**Confidence:** 4

**Summary:**

This work introduces two key contributions: the AesCoT data construction pipeline, which provides high-quality data for SFT training, and the RAPO algorithm, which features a novel dual-reward mechanism comprising a Relative Rank Reward and an Absolute Error Reward, specifically tailored for IAA. However, the overall novelty of this work is limited, and there are certain deficiencies in both the data construction pipeline and the reward design.

**Strengths:**

1. This work supplies high-quality data for aesthetic reasoning by constructing an automated pipeline.
2. It explores the application of reinforcement learning to the IAA domain, demonstrating its viability.
3. Through a series of comprehensive experiments, the study furnishes robust empirical validation for its claims.

**Weaknesses:**

1. The five evaluation dimensions proposed in the AESCOT process have certain limitations and lack support from relevant literature. In practical IAA, the assessment of image aesthetics is more flexible and cannot be comprehensively summarized by the five dimensions suggested by the authors.
2. The filtering strategy mentioned in Section 3.2 and Appendix E is overly simplified, which raises considerable doubts about the credibility of the dataset.
3. The method innovation is limited. Although the study focuses on applying RL to the IAA field, the overall pipeline closely resembles that of DeepSeek-R1, similarly employing high-quality data for SFT cold-start. Moreover, the proposed RAPO and DAPO are highly similar, with modifications to the reward function. The paper leans more toward application rather than introducing substantial innovations.
4. The Relative Rank Reward relies on a critical assumption that the aesthetic score of an image follows a Gaussian distribution. All subsequent reward-rank calculations are based on this assumption, yet no empirical evidence is provided to validate that the scores indeed conform to a Gaussian distribution. The authors should substantiate the correctness of this assumption.
5. Equation 3 essentially models the probability of which image is superior between two images. Why not directly employ the Bradley-Terry model for preference modeling instead?

**Questions:**

Please refer to weakness

---

> ### Author Response · Authors · 2025-11-20
> **Responses to Weakness 1 and Weakness 2**
>
> **`Weakness 1`** : The reliability of the aesthetic dimensions of AesCoT.
>
> ---
>
> 1. **Is there any relevant literature supports the aesthetic dimensions of AesCoT?**
>
> The **Principles of Aesthetics (PoA)** have been discussed in design and art theory literature[1][2][3][4], though exact categorizations vary across sources. Common frameworks identify core principles including balance, harmony, unity, contrast, emphasis, and proportion. For example, the design fundamentals approach articulated by UC Berkeley systematically decomposes visual aesthetics into measurable components—emphasizing how spatial relationships (proportion, balance), visual dynamics (contrast, emphasis), and perceptual cohesion (harmony, unity) collectively determine aesthetic quality.
>
> Our five-dimensional AesCoT framework is derived from common Principles of Aesthetics. Starting from the core principles, we group them into four technical dimensions: Composition (proportion, balance, pattern, unity), Light and Shadow (contrast, emphasis, harmony), Color (harmony and unity), and Exposure (balance and contrast).Beyond these technical dimensions, we specially include Mood and Narrative to extend the framework to capture implicit attributes, which is increasingly recognized as essential to aesthetic evaluation but often overlooked in classical formulations.
>
> [1] Carl Thurston. The ”principles” of art. The Journal of Aesthetics and Art Criticism, 4(2):96–100, 1945.
>
> [2]BBC. “Principles of Design”. https://www.bbc.co.uk/bitesize/topics/zn7cdnb
>
> [3]UC Berkeley. “Design Fundamentals: Elements & Principles”. https://guides.lib.berkeley.edu/c.php?g=920740&p=6634741
>
> [4] Design Fundamentals: Notes on Visual Elements & Principles of Composition
>
> 2. **How to adapt the AesCoT to practical IAA situations?**
>
> We fully agree with your idea that IAA is inherently subjective and highly context-dependent. Relying solely on predefined dimensions may not comprehensively address all specialized aesthetic scenarios (e.g., poster design aesthetics, classical art evaluation, architectural beauty assessment).
>
> On the one hand, our primary focus is on **General Image Aesthetic Assessment (GIAA)**. We designed several universal aesthetic dimensions (like the 3H principles for alignment) as the **structural backbone** mainly for constructing aesthetic reasoning chains, but not to provide a rigid principle to fit with every aesthetic scenario.
>
> On the other hand, the AesCoT construction pipeline we propose is **highly automated, reusable, and customizable.** In practical applications, researchers could seamlessly adapt our framework by redefining or extending aesthetic dimensions tailored to specific domains, and then follow our systematic pipeline to generate high-quality domain-specific aesthetic reasoning data.
>
> **`Weakness 2`** : The filtering strategy details and the credibility of the dataset.
>
> ---
>
> We apologize for the lack of enough details and we have provided the filter process and the prompt template in Appendix B and E  in our revised version.
>
> Our filtering strategy consists of three stages to ensure the quality and credibility of AesCoT : **Score Leakage**, **Reasoning Inconsistency**, and **Factual Error**.
>
> **Score leakage** : Since our pipeline provides the ground truth score to the model before generating critiques, we expect the reasoning chain to flow naturally from image features to the final score, reflecting genuine aesthetic analysis rather than reverse reasoning patterns. To eliminate such cases, we employ GPT-4.1 to evaluate every generated output, producing a binary classification indicating whether there is score leakage.
>
> **Reasoning Inconsistency**: Even powerful closed-source models can occasionally produce judgments that misalign with image quality. This layer targets cases where the generated reasoning chain **contradicts** the GT score. We leverage GPT4.1 and given both the generated reasoning chain and the GT score, the model returns a binary judgment on their consistency.
>
> **Factual error**: Model-generated reasoning chains may suffer from hallucinations, describing non-existent elements or visual features, thereby undermining the validity of the explanations. To address this, we first use GPT-4.1 to generate detailed captions directly from the images. Then we perform an additional filtering to verify whether the content in the reasoning chain aligns with the caption, ensuring factual grounding in actual image content.
>
> **Human Verification**: After these three automated filtering stages, we recruited 5 highly educated students to conduct manual verification of flitered data according to the same three criteria.
>
> In summary, to ensure the reasoning capability the model acquires during the cold-start phase, the AesCoT generation process is both automated and rigorously filtered. We believe these multi-stage quality controls substantially enhance the credibility of AesCoT.

---

> ### Author Response · Authors · 2025-11-20
> **Response to Weakness 3**
>
> **`Weakness 3：`** The innovation of our method
>
> As mentioned in Weakness 5, Traditional approaches in preference learning typically rely on Reward Models (RM) to optimize preference evaluation. However, these methods can only provide **scalar reward signals** and lack the capability to model the **reasoning process**.  In contrast, the emergence of Generative Reward Models (GRM) has introduced new possibilities for preference evaluation tasks by generating intermediate reasoning processes and providing richer learning signals for the model. Our method is more closely aligned with the GRM framework, **emphasizing the modeling and** **optimization** **of evaluation reasoning processes**. A currently popular pipeline is employing RLVR (Reinforcement Learning via Verifiable Rewards). This paradigm has achieved remarkable success in verifiable tasks such as mathematical reasoning and code generation.
>
> **Critically, we do not simply apply this pipeline to the IAA domain.** In our pilot study, we discovered that open-ended evaluation tasks (such as IAA) fundamentally differ from closed-ended tasks like mathematics and coding, which is easy to verify. First, open-ended evaluation tasks often lack carefully annotated high-quality reasoning data, making it difficult for models to acquire sufficient domain **prior knowledge** during the SFT phase, consequently hampering their ability to rollout high-quality results worthy of learning from their distribution. Additionally, for **open-ended verification** tasks, directly using outcome rewards often fails to provide stable and reliable reward signals, while process rewards require dense step-level supervision.
>
> Based on these observations, we first designed an automated high-quality reasoning data construction pipeline. Second, to address the pain points of previous Outcome Reward Model (ORM) methods, we propose **Relative-Absolute Policy** **Optimization** **(RAPO)**, specifically designed for **open-ended, highly subjective** evaluation tasks. RAPO provides stable reward signals through relative-absolute joint modeling, without relying on definitive "correct answers," instead learning human evaluation preference patterns.
>
> We selected IAA as our application scenario because it represents an exceptionally challenging open-ended evaluation task requiring **deep reasoning and professional understanding**, making it ideal for demonstrating the superiority of our approach. More broadly, our work shows that such open-ended evaluation tasks can be successfully situated within the RL scaling paradigm. Our framework **possesses high generalizability** and will be extended to additional open-ended evaluation domains such as creative writing, video quality assessment, and music composition evaluation in future work.

---

> ### Author Response · Authors · 2025-11-20
> **Responses to Weakness 4 and Weakness 5**
>
> **`Weakness 4：`** The assumption that the aesthetic score of an image follows a Gaussian distribution
>
> ---
>
> Empirically, we first visualize the score histograms for each dataset and include these plots in the **Appendix F**, from which one can observe that the score distributions are approximately bell-shaped and close to a normal distribution. Second, we further assess the normality assumption by plotting Q–Q plots and computing the basic statistics and RMSE between the empirical score distribution and its fitted Gaussian distribution (as shown in the table below), which consistently yields small errors, indicating a good fit.
>
> |**Dataset**|**AVA**|**TAD66K**|**FLICKR**|**PARA**|**AADB**|**Combined Train set**|
> |---|---|---|---|---|---|---|
> |**Num_samples**|255502|52248|36449|28220|6958|15000|
> |**Mean**|0.5383|0.5436|0.6029|0.6297|0.6081|0.5560|
> |**Std**|0.0731|0.1447|0.1555|0.1088|0.1525|0.1275|
> |**RMSE**|0.0028|0.0177|0.0185|0.0281|0.0213|0.0118|
>
> From a theoretical perspective, the **Central Limit Theorem**[1] states that, under mild conditions, the sampling distribution of the standardized sample mean converges to a standard normal distribution even when the underlying variables are not normally distributed. Since image aesthetic scores in our setting are obtained as Mean Opinion Scores (MOS) aggregated from multiple independent annotators, this averaging procedure naturally satisfies the conditions of the Central Limit Theorem and supports the Gaussian assumption for MOS.
>
> Moreover, prior work on image quality and aesthetic assessment has also modeled image features, quality scores, or aesthetic scores with Gaussian distributions [2][3]. In particular, the AVA dataset[4] conducted a large-scale Goodness-of-Fit (GoF) study on more than 250k images and reported that, for images with mean scores between 2 and 8 (covering 99.77% of the data), the RMSE of the Gaussian fit is rarely above 0.06, leading to the conclusion that “score distributions are largely Gaussian.”
>
> Based on these studies, our assumption therefore follows both established theoretical justification and widely adopted empirical practice in the image aesthetics literature.
>
>
>
> [1] https://en.wikipedia.org/wiki/Central_limit_theorem
>
> [2] Making a ‘Completely Blind’ Image Quality Analyzer
>
> [3] Source-free Unsupervised Domain Adaptation for Blind Image Quality Assessment
>
> [4] AVA: A Large-Scale Database for Aesthetic Visual Analysis
>
> **`Weakness 5：`** Why not directly employ the Bradley-Terry model for preference modeling instead?
>
> ---
>
> Equation 3 in our paper is actually based on the Thurstone model. The Thurstone model explicitly accounts for uncertainty in preferences by assuming that pairwise preferences follow a normal distribution. Prior research has demonstrated that the essence of BT reward modeling is the **order consistency** [1], BT reward modeling often **overfit to specific patterns** in the training data and are highly sensitive to dataset size and coverage, frequently leading to **reward over-optimization** and **reward hacking** [2][3][4][5][6]. In addition, the Thurstone probabilistic framework naturally aligns with our Gaussian assumption for aesthetic score distributions (as discussed in our response to Weakness 4) .
>
> Empirically, we conduct an ablation experiment and retrain the model using Bradley-Terry model to models the probability between two images with the same settings.
>
> |   |   |   |   |   |   |   |   |   |   |   |   |   |
> |---|---|---|---|---|---|---|---|---|---|---|---|---|
> ||**TAD66K**|  |**AVA**| |**FLICKR**| |**PARA**| |**AADB**| |**Average**| |
> ||**PLCC**|**SRCC**|**PLCC**|**SRCC**|**PLCC**|**SRCC**|**PLCC**|**SRCC**|**PLCC**|**SRCC**|**PLCC**|**SRCC**|
> |RAPO(BT)|0.4093|0.3812|0.6239|0.6148|0.7249|0.7017|0.7710|0.7680|0.5103|0.5020|0.6079|0.5935|
> |RAPO|**0.4147**|**0.3915**|**0.6294**|**0.6242**|**0.7422**|**0.7173**|**0.8192**|**0.7813**|**0.5428**|**0.5369**|**0.6297**|**0.6102**|
>
> It's worth noting in the table above that BT-based model exhibits substantially degraded performance on out-of-distribution (OOD) datasets(PARA,AADB) . In contrast, our proposed Thurstone-based pairwise modeling effectively captures preference uncertainty and generalizes robustly to unseen aesthetic domains, achieving more stable and better performance.
>
> [1] Rethinking Bradley-Terry Models in Preference-Based Reward Modeling: Foundations, Theory, and Alternatives
>
> [2] Think-RM: Enabling Long-Horizon Reasoning in Generative Reward Models
>
> [3] TODO: Enhancing LLM Alignment with Ternary Preferences
>
> [4] Beyond Bradley-Terry Models: A General Preference Model for Language Model Alignment
>
> [5] InfoRM: Mitigating Reward Hacking in RLHF via Information-Theoretic Reward Modeling
>
> [6] Rrm: Robust reward model training mitigates reward hacking
>
>  ---
>
> Finally, we would like to express our sincere gratitude for your active engagement and instructive comments ! We would be more than willing to address any additional concerns you may have.

---

> ### Author Response · Authors · 2025-11-27
> **Thank you & Looking forward to further discussion**
>
> Dear Reviewer zj9p,
>
> We deeply appreciate your dedication to engaging in author-reviewer discussions. Recognizing the existing discrepancies in the understanding of our manuscript, we have outlined your key concerns and our responses for enhanced communication:
>
>  **`The aesthetic dimensions of AesCoT`**  ：We referenced foundational art theory to validate the five aesthetic dimensions of AesCoT and clarified the pipeline's extensibility for domain-specific customization.
>
> **`The filtering strategy details`** We disclosed full details of our three-stage filtering pipeline in our revised version to rigorously ensure data quality and eliminate hallucinations.
>
> **`The innovation of our method`** We differentiated open-ended aesthetic tasks from verifiable tasks (like math/code) to justify why our RAPO framework is uniquely necessary and superior to standard RLVR pipelines in this IAA context.
>
> **`The Gaussian distribution assumption`** We validated the Gaussian distribution assumption of aesthetic scores through extensive empirical goodness-of-fit tests with visualization (Q-Q plots, distribution) in our revised version, as well as theoretical grounding based on the Central Limit Theorem.
>
> **`Thurstone vs. BT Model`** We diligently presented ablation studies and analyzed recent research to demonstrate that our Thurstone-based preference modeling significantly outperforms Bradley-Terry approaches in handling uncertainty and out-of-distribution generalization under the assumption of Gaussian distribution.
>
> For any other issues not mentioned here, please refer to our detailed rebuttal response. We sincerely and humbly believe we have responded to your concerns properly, and we once again thank you for your meticulous and generous review!
>
> Warm regards,
>
> Authors

---

### Author Response · Authors · 2025-12-03
**General response [1/2]**

Dear Program Chairs, Senior Area Chairs, Area Chairs, and Reviewers:

We sincerely thank all ACs, SACs, and PC for your time, selfless dedication and the your commitment to maintaining high standards of the during handling our submission. We also truly value the constructive feedback from all reviewers, which has helped us significantly improve our work. We would like to first restate our core contributions and innovations, followed by a summary of the efforts of key revisions and additional experiments we have incorporated.


 **1. Our core contributions and innovations**

We believe our work provides:
- **A Novel and well-motivated framework for open-ended preference evaluation**:
 We identify and address the limitations of applying standard outcome-based RLVR (designed for verifiable tasks like math/code) to open-ended domains, where mainstream RLVR pipelines with outcome-only rewards often fail to provide reliable reward signals.  We propose **RAPO**, a more general and principled reinforcement learning algorithm. RAPO is explicitly tailored to **open-ended, highly subjective** tasks, jointly optimizing relative pairwise preferences and absolute score calibration to provide **stable and human-aligned** reward signals.
- **The novel and effective application of  reinforcement learning to Image Aesthetic Assessment**:
We instantiate our RAPO algorithm in **Aes-R1**, a reasoning-centric framework that sets a new state-of-the-art in the challenging domain of Image Aesthetic Assessment (IAA), which requires deep understanding and aesthetic reasoning. To our knowledge, this represents the **first** successful application of such a general, reasoning-enhanced preference-evaluation scheme to IAA. Furthermore, the underlying principles of Aes-R1—specifically the combination of chain-of-thought cold-start and relative-absolute joint optimization(RAPO)—are highly **transferable**, offering a promising template for other open-ended tasks such as creative writing and video quality assessment.
- **The First Aesthetic Reasoning Dataset** : We design an end-to-end data pipeline (**AesCoT**) that automatically synthesize **high-quality, multi-dimensional** aesthetic reasoning chains at scale, providing a practical alternative to expensive human expert annotations in data-sparse regimes. We also carefully design a three-stage filtering procedure to ensure the quality of our dataset.
- **State-of-the-art performance and generalization capability**:
We conduct extensive experiments demonstrating that Aes-R1 achieves state-of-the-art performance across multiple IAA benchmarks with limited training samples, and exhibits substantially stronger out-of-distribution generalization than existing baselines. Beyond performance metrics, we provide in-depth mechanistic analysis of the interplay of different optimization objectives and the role of token entropy in cold-start and RL exploration. We also carefully evaluate both the reasoning quality and scoring consistency of our methods.
- **A fully open source framework**:
We will release all the datasets, models, code to facilitate further research, providing the first comprehensive pipeline for image aesthetic assessment.

---

### Author Response · Authors · 2025-12-03
**General response [2/2]**

**2. A summary of our rebuttal phase**

We are encouraged by the recognition of our work that :
- **Reviewer xmgr** and **FLJS** recognize the **core innovations and the novelty** of our proposed RAPO paradigm.
- **Reviewer zj9p** and **XmWM** acknowledge our contribution on the automatic data generation pipeline and high-quality aesthetic reasoning chains.
- **Reviewer zj9p, XmWM** and **xmgr** acknowledge our **effectiveness** of our well-designed Aes-R1 framework applied to IAA domain.
- **Reviewer zj9p,  XmWM** and **FLJS** agree that our comprehensive experiments demonstrate our **state-of-the-art performance and generalization** with limited training data, furnishing robust empirical validation of our method.

Overall, the majority of reviewers recognize the **core contributions and innovations** of this paper, and we have made every effort to conduct extensive new experiments and granular analysis in order to address all raised concerns.

We have responded to each reviewer individually, and we summarize the key points below:

- We restate the **innovation and the main contribution** of our method. Rather than a simple modification of the algorithm, we proposed a more general and principled paradigm tailored for open-ended and highly subjective tasks, enabling human-aligned preference evaluation.  More broadly, our work shows that such open-ended evaluation tasks can be successfully situated within the RL scaling paradigm.
- **```The reward design and interplay```(Reviewer zj9p, Reviewer xmgr)** :  To better validate our reward design , we conducted weighting coefficient ablations (Section 4.3) to analyze the interplay between reward components. We found that relative and absolute rewards exhibit strong complementarity: relying on either in isolation leads to significant performance degradation, whereas their joint optimization provides the comprehensive supervision necessary for reliable and stable learning signals. We also validate our preference modeling design of not directly using BT models by clarifying the limitaion of BT preference modeling. We also modified our framework to compare the BT based modeling with ours. The results show that our proposed pairwise modeling effectively captures preference uncertainty and generalizes robustly to unseen aesthetic domains, achieving more stable and better performance.
- **```Dataset reliability``` (Reviewer zj9p,xmgr)** : To address the concerns of dataset reliability  , we first  provided theoretical justification for our aesthetic dimension design and GPT-4.1 selection, and clarified our detailed three-layer filtering strategy in the revised manuscript. Then we empirically validated the quality of AesCoT data through: (i) human agreement rate comparisons across different data sources ; and (ii) downstream training analysis demonstrating superior accuracy and entropy preservation, validating both data quality and suitability for RL-based exploration.
- **```The Gaussian distribution assumption```(Reviewer zj9p)**: In order to substantiate the correctness of the assumption of Gaussian distribution based preference modeling , we first theoretically quote the Central Limit Theorem to supports the Gaussian assumption for the mean opinion scores and we also visualize all of our train and test datasets distributions and Q-Q plots. We also computed the basic statistics and RMSE between the empirical score distribution and its fitted Gaussian distribution to support our assumption.
- **```Quantitive evaluation of reasoning quality```(Reviewer xmgr)**: To provide a quantitative evaluation of the textual reasoning process , we conducted both human annotations and Model-Based Evaluation. We perform blind comparisons of reasoning chains from five models in human annotators, and we used GPT-5 as an automated evaluator on a larger sample, assessing quality across eight dimensions. The results validates the superior reasoning quality achieved by our Aes-R1 model.
- We also addressed **additional concerns**, including inference latency **(Reviewer XmWM)**, figure clarifications **(Reviewer XmWM)**, and minor typos **(Reviewer xmgr)**, through careful revisions and comprehensive supplementary descriptions and analyses.

We are deeply grateful for the opportunity to engage with the reviewers' insightful feedback, which has been instrumental in refining our manuscript. Although we have not yet received any further responses from the reviewers probably due to policy changes, we have approached this period with the utmost diligence. We have proactively conducted extensive additional experiments, quantitative evaluations, and ablation studies to thoroughly resolve every concern raised.

We respectfully hope all ACs, SACs, and PC to consider the substantial improvements and rigorous validations included in our rebuttal. We sincerely appreciate your time and dedication in handling our paper and look forward to your positive assessment.

Best regards,

The Authors

---

### Meta-Review · Area_Chair_jkgd · 2026-01-12

**Summary:**

The reviewers' concerns are mainly about the five evaluation dimensions proposed, the filtering strategy, innovation is limited, relative rank reward, similar methods adopted by Q-Insight, and more inference time for MLLM-based approach. They have also raised concerns about reasoning quality not quantitatively evaluated, issues on the AesCoT reasoning data, relative influence of the relative and absolute rewards, implementation clarity. The authors have provided a thorough rebuttal to address these issues.

The novelty issues have been raised by different reviewers. Some are concerned that the novelty would be insufficient, while others believe that this work proposes a novel relative-absolute policy optimization (RAPO) algorithm, which is an RL method with a joint reward function that optimizes both absolute scores and relative rankings, thereby reflecting complex human aesthetic judgment. The authors have provided a detailed rebuttal regarding the novelty of the proposed method.

**Reviewer Concerns:**

The reviewers’ concerns focus on the proposed evaluation dimensions, method issues, limited innovation, higher inference time of the MLLM-based approach, unquantified reasoning quality, the created AesCoT reasoning data, the balance between relative and absolute rewards, and implementation clarity.

Almost all concerns have been addressed.

**Reviewer Scores:**

I believe that most reviewers are likely to maintain or raise their scores.

---

### Decision · Program_Chairs · 2026-01-26

Accept (Poster)